# ReCoG: Relational and Compact Context Graph Learning for Few-shot Molecular Property Prediction

Zeyu Wang [1 2 3]  Xin Zheng [4]  Yao Lu [1 3]  Shanqing Yu [1 3]  Qi Xuan [1 3]  Shirui Pan [2]

## Abstract

Few-shot molecular property prediction (FSMPP) is essential in drug discovery and materials design, where high-quality labeled data are often scarce and expensive to obtain. Despite the promising performance of existing methods, especially context-aware methods, they still face two-fold severe challenges with *insufficient structural context modeling* & *redundant auxiliary context learning*, leading to inadequate context graph exploration and ineffective information utilization for effective molecule representation learning. To address these, in this paper, we propose a novel framework by learning on **Re**lational and **C**ompact c**o**ntext **G**raph, named **ReCoG**, to comprehensively exploit the context graph for expressive molecular property prediction. Specifically, the proposed ReCoG contains two core modules: a **(1) cross-property relational learning module** to better model the structural and relational context information, and a **(2) context graph information bottleneck module** to adaptively suppress irrelevant auxiliary signals for compact context information utilization, followed by a detailed theoretical demonstration regarding the importance of joint relational and compact knowledge extraction in context graphs. Extensive experiments on multiple datasets demonstrate that ReCoG consistently outperforms state-of-the-art methods, validating its superiority. Code is available at the repository.

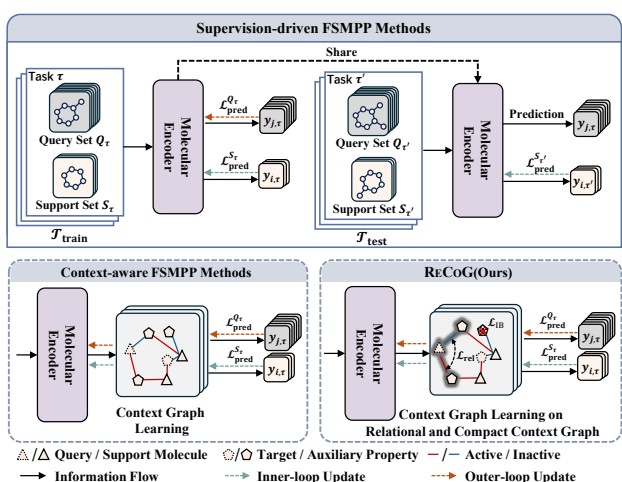

*Figure 1.* The comparison of different FSMPP methods.

## 1. Introduction

Molecular property prediction (MPP) (Walters & Barzilay, 2020), which aims to predict physicochemical and biological properties from molecular structures, is a fundamental task in drug discovery (Bao et al., 2025; Yu et al.) and materials design (Zhou et al., 2025). Recent advances in molecular representation learning (Wang et al., 2024b; Jiang et al., 2024) have substantially improved prediction accuracy with the help of high-quality molecular property annotations as labels. However, in real-world applications, such annotations often rely on expensive and time-consuming experiments or high-fidelity simulations, which makes them difficult to obtain for effective supervised training. To address this label scarcity issue, few-shot molecular property prediction (FSMPP) (Wang et al., 2025b) has emerged to enable rapid adaptation to new molecular properties under limited supervision, where only a small set of labeled molecules is available for the target property.

Early studies (Altae-Tran et al., 2017; Vella & Ebejer, 2022; Guo et al., 2021) on FSMPP primarily combined MPP with the generic few-shot learning (FSL) paradigm, which typically has a small labeled *support* set and an unlabeled *query* set, and optimizes model parameters by minimizing the query objective through bi-level optimization across dif-

[1]Zhejiang University of Technology, Hangzhou, China [2]Griffith University, Gold Coast, Australia [3]Binjiang Institute of Artificial Intelligence, Hangzhou, China [4]Royal Melbourne Institute of Technology, Melbourne, Australia. Correspondence to: Shanqing Yu <yushanqing@zjut.edu.cn>, Shirui Pan <s.pan@griffith.edu.au>.

*Proceedings of the 43rd International Conference on Machine Learning*, Seoul, South Korea. PMLR 306, 2026. Copyright 2026 by the author(s).

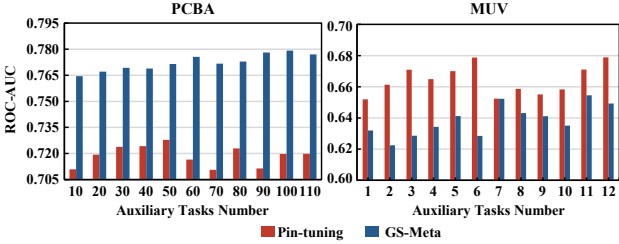

*Figure 2.* The 1-shot results of Pin-Tuning and GS-Meta on PCBA and MUV dataset with different auxiliary task numbers.

ferent molecular learning tasks, as is shown in Figure 1. However, such methods driven solely by limited target-level supervision are usually insufficient to support effective generalization, which treat samples in isolation and fail to exploit dependencies within the cross-task context. Recent studies (Zhuang et al., 2023; Wang et al., 2024a; Li et al., 2025b; Wang et al., 2025a) have begun to incorporate context information, *e.g.*, auxiliary properties and their associations with molecules, by building a context graph that jointly models molecules, target properties, and auxiliary properties, as shown in Figure 1 (left), to develop context-aware FSMPP methods for better adaptation to new tasks.

Despite the substantial progress achieved by existing context-aware methods, they still face two critical challenges: **(C1) Insufficient structural context modeling**. Existing methods mainly incorporate basic auxiliary context information within the context graph, i.e., only captures molecule–property relations, while the relational structure of property-property is not sufficiently modeled, leading to ineffective utilization of context information and hampering the cross-property knowledge transfer. **(C2) Redundant auxiliary context learning**. Existing methods utilize all heterogeneous molecule–property pairs to guide the context-aware few-shot molecular representation learning, while various auxiliary properties inevitably contain task-irrelevant and redundant signals, leading to unstable cross-task generalization. As shown in Figure 2, we examine how the number of auxiliary tasks influences the performance of existing context-aware methods, including Pin-Tuning (Wang et al., 2024a) and GS-Meta (Zhuang et al., 2023), on the PCBA and MUV datasets. The results indicate that increasing the number of auxiliary tasks does not always lead to monotonic performance improvement, and in some cases even degrades learning performance. Given all these challenges and these observations, it naturally leads to two key questions: (1) *How to efficiently exploit the structural information in the context graph?* and (2) *How to eliminate the auxiliary context-induced information redundancy?*

To answer these two questions, in this work, we begin with a comprehensive theoretical analysis of the above challenges, which motivates a novel context graph learning objective

that treats FSMPP as the primary goal while jointly promoting context knowledge extraction and task-irrelevant information filtering. Based on these insights, we propose a novel context graph learning framework for few-shot molecular **Re**lational and **C**ompact c**o**ntext **G**raph, named **ReCoG**, to comprehensively exploit the context graph for expressive molecular property prediction. Specifically, the proposed **ReCoG** has two core modules: **(1) cross-property relation learning module**, which introduces a triplet (i.e., one molecule paired with two properties) property relation learning mechanism to transform implicit context information into optimizable relation signals and facilitate the extraction of informative context knowledge; and **(2) context graph information bottleneck module**, which adaptively filters task-irrelevant context information conditioned on the target property. Extensive experiments demonstrate significant and consistent improvements on multiple FSMPP datasets. The main contributions are summarized as follows:

- We identify two key challenges in FSMPP: *insufficient structural context modeling* & *redundant auxiliary context learning*, and provide a detailed theoretical demonstration of the importance of joint relational and compact knowledge extraction in context graphs.
- We propose a novel framework by learning on a relational and compact context graph, named **ReCoG**, which contains a ***cross-property relation learning module*** and a ***context graph information bottleneck module***, to comprehensively exploit the context graph for expressive molecular property prediction.
- Extensive experiments on multiple few-shot molecular property prediction datasets demonstrate the superiority of **ReCoG** across 10-shot and 1-shot.

## 2. Related Work

Few-shot molecular property prediction (FSMPP) aims to adapt predictive models to novel molecular property tasks with only a few labeled molecules (Wang et al., 2025b). Early works mainly combine molecular property prediction with generic few-shot learning paradigms, such as metric-based methods (Altae-Tran et al., 2017) and meta-learning (Altae-Tran et al., 2017), relying on limited target-task supervision to improve model generalization. However, such supervision alone is often insufficient for robust generalization in FSMPP. To mitigate this issue, recent studies introduce context-aware learning by incorporating external knowledge or exploiting context information within datasets. Representative approaches leverage external molecular databases (Schimunek et al., 2023), auxiliary molecular descriptors (Lv et al., 2023), large language models (Fifty et al., 2023), or construct context graphs to model molecule–molecule and molecule–property relations (Wang et al., 2021; Zhuang et al., 2023; Wang et al.,

2024a; 2025a). These methods form a dominant FSMPP paradigm that combines pre-trained molecular encoders with context-aware prediction heads. Despite their effectiveness, existing context-aware methods primarily focus on modeling explicit context signals and largely overlook implicit cross-property relations that are crucial for knowledge transfer. Moreover, they often indiscriminately incorporate auxiliary context, introducing redundancy that may impair model generalization. These limitations highlight the need for developing a *relational* and *compact* context graphs learning mechanism, which efficiently model cross-property relation and compress task-irrelevant context information.

## 3. Preliminaries

**Notations.** Let $\mathcal{T}$ denote the set of molecular property prediction tasks, where each task $\tau \in \mathcal{T}$ corresponds to a specific property. The training tasks are denoted by $\mathcal{T}_{\text{train}}$, with the associated dataset $\mathcal{D}_{\text{train}} = \{(m_i, y_{i,\tau}) \mid \tau \in \mathcal{T}_{\text{train}}\}$, where $m_i$ denotes a molecule and $y_{i,\tau}$ is its label under task $\tau$. The test set can be denoted as $\mathcal{D}_{\text{test}} = \{(m_j, y_{j,\tau}) \mid \tau \in \mathcal{T}_{\text{test}}\}$. The properties associated with training and testing tasks are assumed to be disjoint, i.e., $\{\mathcal{T}_{\text{train}}\} \cap \{\mathcal{T}_{\text{test}}\} = \varnothing$.

**Problem Definition of FSMPP.** The objective of FSMPP is to learn a predictive model from $\mathcal{D}_{\text{train}}$ that can rapidly adapt to unseen tasks in $\mathcal{T}_{\text{test}}$ using only limited labeled molecules per task. Following prior work, we adopt the episodic training paradigm commonly used in meta-learning, where a batch of episodes $\{E_\tau\}^B$ is iteratively sampled. For each episode $E_\tau$, a target task $\tau \in \mathcal{T}_{\text{train}}$ is first sampled, with a support set $S_\tau$ and a query set $Q_\tau$. In general, each task is formulated as a binary classification problem, i.e., active ($y = 1$) or inactive ($y = 0$). A 2-way $K$-shot episode is constructed such that the support set $S_\tau = \{(m_i, y_{i,\tau})\}_{i=1}^{2K}$ contains $K$ labeled molecules per class, while the query set $Q_\tau = \{(m_i, y_{i,\tau})\}_{i=1}^{M}$ consists of $M$ molecules.

**Context Graph Construction in FSMPP.** Given an episode $E_\tau = (S_\tau, Q_\tau)$, one can construct a context graph $\mathcal{G}_\tau = (\mathcal{V}_\tau, \mathcal{E}_\tau)$ to explicitly model context relations between molecules and properties. The node set consists of molecular nodes and task nodes, i.e., $\mathcal{V}_\tau = \mathcal{V}_\tau^m \cup \mathcal{V}_\tau^p$, where $\mathcal{V}_\tau^m$ represents the set of molecular nodes corresponding to all molecules in the episode, and $\mathcal{V}_\tau^p = \{\tau \cup \{\tau'\}^{N_{\text{auxi}}}\}$ denotes the set of task nodes associated with a target task $\tau$ and $N_{\text{auxi}}$ auxiliary tasks. The edge set $\mathcal{E}_\tau$ encodes the molecule-property relations (i.e., active, inactive, unknown) by connecting molecular nodes to task nodes, i.e., $\mathcal{E}_\tau \subseteq \mathcal{V}_\tau^m \times \mathcal{V}_\tau^p$.

**Information Bottleneck.** The IB principle, originally introduced by (Tishby et al., 1999), provides a theoretical framework for learning task-relevant representations under an explicit compression constraint. Given an input variable $X$, a target variable $Y$, and a latent representation $Z$, the IB

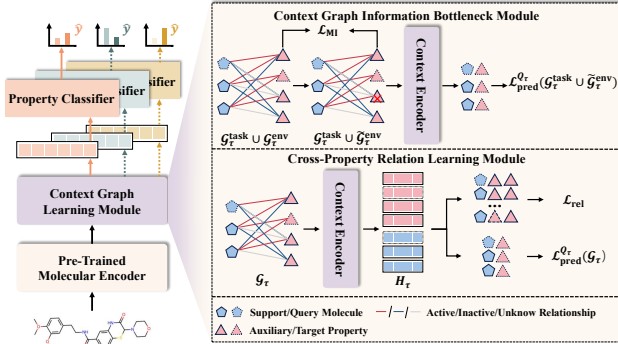

*Figure 3.* The framework of our proposed RECOG.

objective is formulated as:

$$\max_{p(z|x)} \ I(Z; Y) - \beta I(Z; X), \qquad (1)$$

where $I(\cdot; \cdot)$ denotes mutual information (MI) and $\beta > 0$ controls the trade-off parameter. This objective encourages $Z$ to retain information that is predictive of $Y$ while discarding redundant or task-irrelevant information from $X$. As a result, the learned representation achieves a balance between predictive sufficiency and representational compactness.

## 4. Methodology

This section details RECOG, including its theoretical motivation and technical design. RECOG is developed to address two key challenges in existing context-aware FSMPP methods: *insufficient structural context modeling* & *redundant auxiliary context learning*, corresponding to two key components: a cross-property relation learning module and a context graph information bottleneck module. In the following, we first rethink the context graph learning objective in FSMPP to provide theoretical foundations for the importance of joint relational and compact knowledge extraction in context graphs. Then, we provide the details of our proposed framework and optimization objectives. The overall framework is illustrated in Figure 3.

### 4.1. Optimization Rethinking

Given a molecule $m$, the objective of MPP is to learn a mapping $\mathcal{F}_{\text{mol}} : m \to X$ to obtain its representation. We assume the existence of an ideal but unobservable latent semantic variable set $\mathcal{M}$, which characterizes the intrinsic semantics of a molecule in different properties dimensions. The molecular structure $m$ and its associated property labels can be viewed as different observations generated from $\mathcal{M}$. In early FSMPP studies, they only consider a target property $\tau$, the model has access solely to the corresponding label $Y_\tau$ in the training stage. This label can be viewed as a projection of $\mathcal{M}$ onto the $\tau$-th property. The resulting information flow

can be expressed as:

$$Y_\tau \leftarrow \mathcal{M} \rightarrow m \rightarrow X, \qquad (2)$$

and the learning objective is typically formulated as

$$\max I(X; Y_\tau), \qquad (3)$$

which encourages the learned representation to capture the components of $\mathcal{M}$ relevant to the target property under limited supervision.

With the development of context-aware few-shot molecular property prediction methods (Wang et al., 2021; Zhuang et al., 2023; Wang et al., 2025a), given a context graph $\mathcal{G}_\tau$ for a target property $\tau$, most existing methods focus on learning powerful molecular representations by maximizing the mutual information between the context graph and its target property label $Y_\tau$ as:

$$\max I(\mathcal{G}_\tau; Y_\tau). \qquad (4)$$

In this process, model optimization mainly relies on the limited target supervision $Y_\tau$, leaving the latent supervised signals embedded in the context graph underexplored. This naturally leads to two key questions:

**Q1**: How to efficiently exploit the structural information in the context graph?

**Q2**: How to eliminate the auxiliary context-induced information redundancy?

**Optimization Objective Reformulation for Q1.** Within the context graph $\mathcal{G}_\tau$, the model can observe not only the limited supervision $Y_\tau$ but also the auxiliary properties $\{Y_{\tau'}\}_{\tau'=1}^{N_{\text{auxi}}}$. These auxiliary labels induce cross-property relation signals $Y_{\text{rel}}$ (e.g., derived from joint property annotations), denoted as $(Y_\tau, \{Y_{\tau'}\}_{\tau'=1}^{N_{\text{auxi}}}) \rightarrow Y_{\text{rel}}$. From a generative perspective, these signals originate from the same latent semantic space, implying that jointly modeling the target label $Y_\tau$, the auxiliary information $\{Y_{\tau'}\}_{\tau'=1}^{N_{\text{auxi}}}$ and the relation signal $Y_{\text{rel}}$ facilitates a closer approximation to optimal latent representation learning. Motivated by this insight, we propose a cross-property relation learning module. Specifically, we focus on the $Y_{\text{rel}}$ induced by the context that shares semantic relevance with the target task $\tau$. Consequently, we extend the IB objective in Eq. (4) to exploit shared semantics across properties as:

$$\max I[\mathcal{G}_\tau; (Y_\tau, Y_{\text{rel}})]. \qquad (5)$$

According to the mutual information chain rule, the Eq. (5) can be reformulated as:

$$\max \underbrace{I(\mathcal{G}_\tau; Y_{\text{rel}}|Y_\tau)}_{\text{Auxiliary}} + \underbrace{I(\mathcal{G}_\tau; Y_\tau)}_{\text{Target}}, \qquad (6)$$

where $I(\mathcal{G}_\tau; Y_{\text{rel}} \mid Y_\tau)$ incentivizes the learning context graph by capturing auxiliary cross-property dependencies

conditional on the target task, while $I(\mathcal{G}_\tau; Y_\tau)$ maximizes the mutual information between the context graph and the target property to ensure accurate prediction.

**Optimization Objective Reformulation for Q2.** Given a context graph for facilitating FSMPP task with various auxiliary property information, it inevitably introduces task-irrelevant signals for certain target property. In this case, we develop a context graph information bottleneck objective, which decomposes the context graph $\mathcal{G}_\tau = \mathcal{G}_\tau^{\text{task}} \cup \mathcal{G}_\tau^{\text{env}}$, where $\mathcal{G}_\tau^{\text{task}}$ consists of molecular nodes $\mathcal{V}_\tau^m$ and the target task node $\tau \in \mathcal{V}_\tau^p$, and $\mathcal{G}_\tau^{\text{env}}$ is formed by molecular nodes $\mathcal{V}_\tau^m$ and auxiliary property nodes $\{\tau'\}^{N_{\text{auxi}}} \subset \mathcal{V}_\tau^p$. Our goal is to learn a $\widetilde{\mathcal{G}}_\tau^{\text{env}}$ of $\mathcal{G}_\tau^{\text{env}}$ conditioned on $\mathcal{G}_\tau^{\text{task}}$. Thus, the target term of Eq. (6) can be reformulated as:

$$\max I(\mathcal{G}_\tau^{\text{task}} \cup \widetilde{\mathcal{G}}_\tau^{\text{env}}; Y_\tau) - \beta I(\widetilde{\mathcal{G}}_\tau^{\text{env}}, \mathcal{G}_\tau^{\text{env}}|\mathcal{G}_\tau^{\text{task}}). \qquad (7)$$

Together, these objectives yield a unified objective:

$$\max \underbrace{I(\mathcal{G}_\tau; Y_{\text{rel}}|Y_\tau)}_{\text{Auxiliary}} + \underbrace{I(\mathcal{G}_\tau^{\text{task}} \cup \widetilde{\mathcal{G}}_\tau^{\text{env}}; Y_\tau)}_{\text{Target}}$$
$$- \underbrace{\beta I(\widetilde{\mathcal{G}}_\tau^{\text{env}}; \mathcal{G}_\tau^{\text{env}}|\mathcal{G}_\tau^{\text{task}})}_{\text{Compact}}, \qquad (8)$$

which explicitly balances the exploitation of informative context semantics and the suppression of context-induced redundancy. This unified objective highlights that effective context-aware FSMPP requires not only incorporating auxiliary information, but also selectively filtering it.

### 4.2. Model Architecture

We implement ReCoG based on the typical context-aware FSMPP framework that combines a pre-trained molecular encoder with context-aware prediction head. Given an episode $E_\tau = (S_\tau, Q_\tau)$, one can construct a context graph $\mathcal{G}_\tau = (\mathcal{V}_\tau, \mathcal{E}_\tau)$. $\boldsymbol{X} \in \mathbb{R}^{|\mathcal{V}_\tau| \times d_1}$ denotes the node feature matrix, where each node feature is defined in a unified form:

$$\boldsymbol{x}_v = \begin{cases} \mathcal{F}_{\text{mol}}(m_v) \in \mathbb{R}^{d_1}, & \text{if } v \in \mathcal{V}_\tau^m, \\ \boldsymbol{e}_v \in \mathbb{R}^{d_1}, & \text{if } v \in \mathcal{V}_\tau^p, \end{cases} \qquad (9)$$

where $\mathcal{F}_{\text{mol}}(\cdot)$ denotes the pre-trained molecular encoder to map the input molecule to the latent representations, $\boldsymbol{e}_v$ is a learnable embedding vector for the property node, randomly initialized and jointly optimized during training. Given the context graph $\mathcal{G}_\tau$ and the initial node features $\boldsymbol{X}$, one can employ a GNN-based context encoder ContextEncoder$(\cdot)$ to capture context dependencies between molecules and properties: $\boldsymbol{H} = \text{ContextEncoder}(\mathcal{G}_\tau, \boldsymbol{X})$, where $\boldsymbol{H} = \{\boldsymbol{h}_v \in \mathbb{R}^{d_2} \mid v \in \mathcal{V}_\tau\}$. During prediction, for each molecule $m_i$ and target task $\tau$, we directly concatenate their context representations, $\boldsymbol{h}_i$ and $\boldsymbol{h}_\tau$, and feed them into a $N_{\text{pred}}$-layer MLP classifier $\mathcal{F}_{\text{pred}}(\cdot)$ to obtain the final prediction:

$$\hat{y}_{i,\tau} = \mathcal{F}_{\text{pred}}([\boldsymbol{h}_i \,\|\, \boldsymbol{h}_\tau]), \qquad (10)$$

where $[\cdot\|\cdot]$ denotes concatenation, and $\hat{y}_{i,\tau}$ represents the prediction of molecule $m_i$ on property $\tau$. Details about the model configuration are provided in Appendix. C.3.

## 4.3. Model Optimization

### 4.3.1. MAXIMIZING $I(\mathcal{G}_\tau; Y_{\mathrm{rel}} \mid Y_\tau)$

The auxiliary term in Eq. (8) is designed to capture informative context semantics to support target property prediction by modeling cross-property relations. This conditional mutual information admits the following lower bound:

$$I(\mathcal{G}_\tau; Y_{\mathrm{rel}} \mid Y_\tau) \geq \mathbb{E}_{\mathcal{G}_\tau, Y_\tau, Y_{\mathrm{rel}}}\Big[ \log q_\phi(Y_{\mathrm{rel}} \mid \mathcal{G}_\tau) \Big] \\ + H(Y_{\mathrm{rel}} \mid Y_\tau). \quad (11)$$

The detailed proof is provided in Appendix. A.1. Since the conditional entropy term $H(Y_{\mathrm{rel}} \mid Y_\tau)$ is independent of model parameters, it can be treated as a constant and omitted during optimization. The expectation $\mathbb{E}[\log q_\phi(Y_{\mathrm{rel}} \mid \mathcal{G}_\tau)]$ encourages the context graph to exploit relational semantics shared across different properties. As such cross-property relations are not directly observable, we model $Y_{\mathrm{rel}}$ from an implicit relation signal derived from molecular labels under different properties. Specifically, for molecule $m_i$ and two properties $\tau_1, \tau_2$, we define:

$$y_{\mathrm{rel}}^{(i,\tau_1,\tau_2)} \triangleq (y_{i,\tau_1} - y_{i,\tau_2})^2, \quad (12)$$

which is non-negative and invariant to properties orders. Particularly, for molecules $m_i \in S_\tau$, $\tau_1, \tau_2 \in \{\tau \cup \{\tau'\}^{N_{\mathrm{auxi}}}\}$, for molecules $m_i \in Q_\tau$, one can only get their auxiliary tasks labels, i.e., $\tau_1, \tau_2 \in \{\tau'\}^{N_{\mathrm{auxi}}}$. Let $\mu_\phi(\mathcal{G}_\tau) \in \mathbb{R}$ denote the output of a relational prediction head. We assume a Gaussian conditional likelihood for the relation signal:

$$q_\phi(Y_{\mathrm{rel}} \mid \mathcal{G}_\tau) = \mathcal{N}\big(Y_{\mathrm{rel}}; \mu_\phi(\mathcal{G}_\tau), \sigma_{\mathrm{rel}}^2\big), \quad (13)$$

where $\sigma_{\mathrm{rel}}^2$ is a fixed variance. The corresponding log-likelihood is:

$$\log q_\phi(Y_{\mathrm{rel}} \mid \mathcal{G}_\tau) = -\frac{1}{2\sigma_{\mathrm{rel}}^2}(Y_{\mathrm{rel}} - \mu_\phi(\mathcal{G}_\tau))^2 - \frac{1}{2}\log(2\pi\sigma_{\mathrm{rel}}^2). \quad (14)$$

Since the constant term $-\frac{1}{2}\log(2\pi\sigma_{\mathrm{rel}}^2)$ does not affect optimization, maximizing the expected log-likelihood is equivalent to minimizing the squared error:

$$\mathcal{L}_{\mathrm{rel}}^{\mathrm{reg}} \propto \mathbb{E}\Big[\big(Y_{\mathrm{rel}} - \mu_\phi(\mathcal{G}_\tau)\big)^2\Big]. \quad (15)$$

In practice, the relational mean is implemented as:

$$\hat{y}_{\mathrm{rel}}^{(i,\tau_1,\tau_2)} = \mathcal{F}_{\mathrm{rel}}\Big([\boldsymbol{h}_i\|\boldsymbol{h}_{\tau_1}] \odot [\boldsymbol{h}_i\|\boldsymbol{h}_{\tau_2}]\Big), \quad (16)$$

where $\mathcal{F}_{\mathrm{rel}}$ denotes $N_{\mathrm{rel}}$-layer MLP relation classifier, $\boldsymbol{h}_i$ denotes the context representation of molecule $m_i$ extracted

from $\mathcal{G}_\tau$, and $\boldsymbol{h}_{\tau_1}$ and $\boldsymbol{h}_{\tau_2}$ are the corresponding context representations of task nodes. The final relational learning loss is defined as:

$$\mathcal{L}_{\mathrm{rel}} = \frac{1}{2}\Bigg[ \sum_{m_i \in S_\tau} \sum_{\tau_1,\tau_2 \in \mathcal{V}^p} \mathrm{MSE}\Big(\hat{y}_{\mathrm{rel}}^{(i,\tau_1,\tau_2)}, y_{\mathrm{rel}}^{(i,\tau_1,\tau_2)}\Big) \\ + \sum_{m_i \in Q_\tau} \sum_{\tau_1,\tau_2 \in \mathcal{V}^p \backslash \tau} \mathrm{MSE}\Big(\hat{y}_{\mathrm{rel}}^{(i,\tau_1,\tau_2)}, y_{\mathrm{rel}}^{(i,\tau_1,\tau_2)}\Big) \Bigg]. \quad (17)$$

### 4.3.2. MAXIMIZING $I(\mathcal{G}_\tau^{\mathrm{task}} \cup \widetilde{\mathcal{G}}_\tau^{\mathrm{env}}; Y_\tau)$

The objective of the target term in Eq. (8) is to preserve task-relevant information within the refined context graph $\mathcal{G}_\tau^{\mathrm{task}} \cup \widetilde{\mathcal{G}}_\tau^{\mathrm{env}}$. Specifically, $I(\mathcal{G}_\tau^{\mathrm{task}} \cup \widetilde{\mathcal{G}}_\tau^{\mathrm{env}}; Y_\tau)$ can be lower-bounded by:

$$I(\mathcal{G}_\tau^{\mathrm{task}} \cup \widetilde{\mathcal{G}}_\tau^{\mathrm{env}}; Y_\tau) \\ \geq \mathbb{E}_{\mathcal{G}_\tau^{\mathrm{task}}, \widetilde{\mathcal{G}}_\tau^{\mathrm{env}}, Y_\tau}\Big[\log q_\theta(Y_\tau \mid \mathcal{G}_\tau^{\mathrm{task}} \cup \widetilde{\mathcal{G}}_\tau^{\mathrm{env}})\Big] + H(Y_\tau), \quad (18)$$

where $q_\theta(Y_\tau \mid \mathcal{G}_\tau^{\mathrm{task}} \cup \widetilde{\mathcal{G}}_\tau^{\mathrm{env}})$ denotes a variational approximation to the true posterior. The marginal entropy term $H(Y_\tau)$ depends only on the data distribution and is independent of the model parameters; thus, it can be treated as a constant. Consequently, maximizing the above lower bound reduces to maximizing the expected log-likelihood term, which is equivalent to minimizing the negative log-likelihood of the predictor. In practice, we parameterize $q_\theta(\cdot)$ using a task-specific prediction head that takes the refined context graph as input. Therefore, maximizing the lower bound of $I(\mathcal{G}_\tau^{\mathrm{task}} \cup \widetilde{\mathcal{G}}_\tau^{\mathrm{env}}; Y_\tau)$ can be directly achieved by minimizing the target task prediction loss $\mathcal{L}_{\mathrm{pred}}(Y_\tau \mid \mathcal{G}_\tau^{\mathrm{task}} \cup \widetilde{\mathcal{G}}_\tau^{\mathrm{env}})$. A detailed derivation of the lower bound is provided in Appendix. A.2.

### 4.3.3. MINIMIZING $I(\widetilde{\mathcal{G}}_\tau^{\mathrm{env}}; \mathcal{G}_\tau^{\mathrm{env}} | \mathcal{G}_\tau^{\mathrm{task}})$

To effectively filter out superfluous information in the subgraph $\mathcal{G}_\tau^{\mathrm{env}}$, we aim to minimize the conditional mutual information $I(\widetilde{\mathcal{G}}_\tau^{\mathrm{env}}; \mathcal{G}_\tau^{\mathrm{env}} | \mathcal{G}_\tau^{\mathrm{task}})$. By applying the mutual information chain rule, we decompose this term as:

$$I(\widetilde{\mathcal{G}}_\tau^{\mathrm{env}}; \mathcal{G}_\tau^{\mathrm{env}} | \mathcal{G}_\tau^{\mathrm{task}}) = I(\widetilde{\mathcal{G}}_\tau^{\mathrm{env}}; \mathcal{G}_\tau^{\mathrm{env}}, \mathcal{G}_\tau^{\mathrm{task}}) - I(\widetilde{\mathcal{G}}_\tau^{\mathrm{env}}; \mathcal{G}_\tau^{\mathrm{task}}). \quad (19)$$

In principle, one possible way to minimize the conditional mutual information is to simultaneously minimize $I(\widetilde{\mathcal{G}}_\tau^{\mathrm{env}}; \mathcal{G}_\tau^{\mathrm{env}}, \mathcal{G}_\tau^{\mathrm{task}})$ and maximize $I(\widetilde{\mathcal{G}}_\tau^{\mathrm{env}}; \mathcal{G}_\tau^{\mathrm{task}})$. However, empirically, we observe that explicitly maximizing this term leads to a degradation in model performance (See Section D.1). We attribute this phenomenon to the *complementary* design of the context graph $\mathcal{G}_\tau$: subgraph $\mathcal{G}_\tau^{\mathrm{task}}$ is intended to capture essential task structure, while subgraph $\mathcal{G}_\tau^{\mathrm{env}}$ encodes auxiliary context patterns. Forcibly increasing their mutual information causes the auxiliary semantics

to redundantly align with the essential semantics, thereby undermining the model's capacity to preserve diverse context semantics. Consequently, we omit the second term and focus on minimizing $I(\widetilde{\mathcal{G}}_\tau^{\text{env}}; \mathcal{G}_\tau^{\text{env}}, \mathcal{G}_\tau^{\text{task}})$.

To achieve this information compression, we adopt a strategy that injects noise into the node features to encourage the encoder to discard the superfluous information (Wu et al., 2020; Lee et al., 2023). Specifically, given the initial node features $\boldsymbol{X}^{\text{env}}$ of subgraph $\mathcal{G}_\tau^{\text{env}}$, we generate a probability mask to control the information flow. For each auxiliary task node $\tau'$ in $\mathcal{G}_\tau^{\text{env}}$, we calculate a probability $p_{\tau'}$ with the context graph node features:

$$p_{\tau'} = \text{MLP}\big([x_{\tau'}^{\text{env}} || z_{\mathcal{G}_\tau^{\text{task}}}]\big), \tag{20}$$

where $z_{\mathcal{G}_\tau^{\text{task}}}$ denotes the subgraph representation of $\mathcal{G}_\tau$, i.e., $z_{\mathcal{G}_\tau^{\text{task}}} = \text{MeanPooling}(\boldsymbol{X}^{\text{task}})$. A Bernoulli gate $\lambda_{\tau'} \sim \text{Bernoulli}(p_{\tau'})$ is relaxed via the Gumbel-Sigmoid trick (Maddison et al., 2017) to enable backpropagation through discrete sampling:

$$\lambda_{\tau'} = \sigma\big(\frac{1}{t}(\log\frac{p_{\tau'}}{1-p_{\tau'}} + \log\frac{u}{1-u})\big), u \sim \text{Uniform}(0,1), \tag{21}$$

where $t$ is the temperature hyperparameter. Based on the $\lambda_{\tau'}$, one can get a perturbed representation $\widetilde{\boldsymbol{X}}_{\tau'}^{\text{env}}$:

$$\widetilde{\boldsymbol{X}}_{\tau'}^{\text{env}} = \lambda_{\tau'}\boldsymbol{X}_{\tau'}^{\text{env}} + (1-\lambda_{\tau'})\epsilon, \quad \epsilon \sim \mathcal{N}(\mu_{\boldsymbol{X}^{\text{env}}}, \sigma_{\boldsymbol{X}^{\text{env}}}^2), \tag{22}$$

where $\mu_{\boldsymbol{X}^{\text{env}}}$ and $\sigma_{\boldsymbol{X}^{\text{env}}}^2$ are the mean and variance of the current batch embeddings, and $\lambda_{\tau'}$ acts as a gate drawn from a Bernoulli distribution parameterized by $p_{\tau'}$. Finally, we minimize the variational upper bound of the mutual information, which acts as a regularization loss to compress the input graph information into $\widetilde{\mathcal{G}}_\tau^{\text{env}}$:

$$\begin{aligned} I(\widetilde{\mathcal{G}}_\tau^{\text{env}}; &\mathcal{G}_\tau^{\text{env}}, \mathcal{G}_\tau^{\text{task}}) \\ &\leq \mathbb{E}_{\mathcal{G}_\tau^{\text{env}}, \mathcal{G}_\tau^{\text{task}}}\left[-\frac{d_1}{2}\log A + \frac{d_1}{2N_{\text{auxi}}}A + \frac{1}{2N_{\text{auxi}}}\|B\|_2^2\right] \\ &:= \mathcal{L}_{\text{MI}}, \end{aligned} \tag{23}$$

where $A = \sum_{\tau'=1}^{N_{\text{auxi}}}(1-\lambda_{\tau'})^2$ quantifies the injected noise, and $B = \frac{\sum_{\tau'=1}^{N_{\text{auxi}}}\lambda_{\tau'}(\boldsymbol{X}_{\tau'}^{\text{env}} - \mu_{\boldsymbol{X}^{\text{env}}})}{\sigma_{\boldsymbol{X}^{\text{env}}}}$ represents the normalized deviation of the retained features. By minimizing $\mathcal{L}_{\text{MI}}$, the model learns to filter out task-irrelevant information from $\mathcal{G}_\tau^{\text{env}}$ while preserving the necessary context compatible with the $\mathcal{G}_\tau^{\text{task}}$. The proof is provided in Section A.3.

### 4.3.4. BI-LEVEL OPTIMIZATION

Following the optimization mechanism of MAML, we adopt a bi-level gradient descent strategy, including the inner-loop and outer-loop. Our ReCoG mainly focus on the optimization of outer-loop. Specifically, given the FSMPP model

with parameter $\zeta$, a batch of $B$ episodes $\{E_\tau\}_{\tau=1}^B$ is randomly sampled from the training distribution. In the inner loop, for each episode $E_\tau$, the model adapts to the specific task using the support set $S_\tau$. The adaptation is performed by minimizing the prediction loss with the supervision from support set $\mathcal{L}_{\text{pred}}^{S_\tau}(Y_\tau|\mathcal{G}_\tau)$ via gradient descent:

$$\begin{aligned} \mathcal{L}_{\text{pred}}^{S_\tau} &= \mathbb{E}_{\mathcal{G}_\tau, Y_\tau}\big[(-\log q_\theta(Y_\tau|\mathcal{G}_\tau)]\big] \\ &= -\sum_{m_i \in \mathcal{S}_\tau}(y_{i,\tau}\log(\hat{y}_{i,\tau}) + (1-y_{i,\tau})\log(1-\hat{y}_{i,\tau})), \end{aligned} \tag{24}$$

$$\zeta' \leftarrow \zeta - \alpha_{\text{inner}}\nabla_\zeta\mathcal{L}_{\text{pred}}^{S_\tau}, \tag{25}$$

where $\alpha_{\text{inner}}$ is the inner-loop learning rate, and $\zeta_\tau'$ represents the task-specific parameters adapted to episode $E_\tau$. In outer-loop, we train the model with all modules:

$$\begin{aligned} \mathcal{L}_{\text{total}}^{Q_\tau} =& \mathcal{L}_{\text{pred}}^{Q_\tau}(Y_\tau|\mathcal{G}_\tau) + \mathcal{L}_{\text{rel}} + \beta\mathcal{L}_{\text{MI}} \\ &+ \mathcal{L}_{\text{pred}}^{Q_\tau}(Y_\tau|\mathcal{G}_\tau^{\text{task}} \cup \hat{\mathcal{G}}_\tau^{\text{env}}), \end{aligned} \tag{26}$$

$$\zeta \leftarrow \zeta - \alpha_{\text{outer}}\nabla_\zeta\big(\frac{1}{B}\sum_{\tau=1}^B\mathcal{L}_{\text{total}}^{Q_\tau}\big), \tag{27}$$

where $\beta$ is a hyperparameter that controls context information compression, and $\alpha_{\text{outer}}$ denotes the learning rate of outer-loop. Pseudo code is provided in Appendix. B.

## 5. Experiments

### 5.1. Experiment Setup

**Datasets.** Experiments are conducted on five benchmark datasets from MoleculeNet (Wu et al., 2018): Tox21, SIDER, MUV, ToxCast, and PCBA, which are widely used for few-shot molecular property prediction. We adopt the same FSMPP data splits as prior methods (Zhuang et al., 2023) for fair comparison. Additional dataset details and statistics are reported in Appendix. C.1.

**Baselines.** We compare our method with a diverse set of representative few-shot molecular property prediction baselines, covering both training-from-scratch and pretrained-encoder paradigms. Specifically, the first group consists of methods that train molecular encoders from scratch, including Siamese Network (Koch et al., 2015), ProtoNet (Snell et al., 2017), MAML (Finn et al., 2017), TPN (Liu et al., 2019), EGNN (Kim et al., 2019), IterRefLSTM (Altae-Tran et al., 2017), and MHNfs (Schimunek et al., 2023). The second group includes methods that leverage pretrained molecular encoders, namely Pre-GNN (Hu et al., 2020), Meta-MGNN (Guo et al., 2021), Pre-PAR (Wang et al., 2021), Pre-GS-Meta (Zhuang et al., 2023), Pre-ADKF-IFT (Chen et al., 2023), Pre-PACIA (Wu et al., 2024), Pre-KRGTS (Wang et al., 2025a), Pin-Tuning (Wang et al., 2024a), and Uni-Match (Li et al., 2025a). More details and implementations are provided in Appendix. C.2.

*Table 1.* Few-shot molecular property prediction performance (ROC-AUC %) on five benchmark datasets. Boldface and underline denote the best and sub-optimal results, respectively, while '–' indicates unreported results.

| Method | Tox21 | | SIDER | | MUV | | ToxCast | | PCBA | |
|---|---|---|---|---|---|---|---|---|---|---|
| | 10-shot | 1-shot | 10-shot | 1-shot | 10-shot | 1-shot | 10-shot | 1-shot | 10-shot | 1-shot |
| Siamese | 80.40±0.35 | 65.00±1.58 | 71.10±4.32 | 51.43±3.31 | 59.96±5.13 | 50.00±0.17 | – | – | – | – |
| ProtoNet | 74.98±0.32 | 65.58±1.72 | 64.54±0.89 | 57.50±2.34 | 65.88±4.11 | 58.31±3.18 | 63.70±1.26 | 56.36±1.54 | 64.93±1.94 | 55.79±1.45 |
| MAML | 80.21±0.24 | 75.74±0.48 | 70.43±0.76 | 67.81±1.12 | 63.90±2.28 | 60.51±3.12 | 66.79±0.85 | 65.97±5.04 | 66.22±1.31 | 62.04±1.73 |
| TPN | 76.05±0.24 | 60.16±1.18 | 67.84±0.95 | 62.90±1.38 | 65.22±5.82 | 50.00±0.51 | 62.74±1.45 | 50.01±0.05 | – | – |
| EGNN | 81.21±0.16 | 79.44±0.22 | 72.87±0.73 | 70.79±0.95 | 65.20±2.08 | 62.18±1.76 | 63.65±1.57 | 61.02±1.94 | 69.92±1.85 | 62.14±1.58 |
| IterRefLSTM | 81.10±0.17 | 80.97±0.10 | 69.63±0.31 | 71.73±0.14 | 49.56±5.12 | 48.54±3.12 | – | – | – | – |
| MHNfs | 80.23±0.84 | – | 65.89±1.17 | – | 73.81±2.53 | – | 74.91±0.73 | – | – | – |
| Pre-GNN | 82.14±0.08 | 81.68±0.09 | 73.96±0.08 | 73.24±0.12 | 67.14±1.58 | 64.51±1.45 | 73.68±0.74 | 72.90±0.84 | 76.79±0.45 | 75.27±0.49 |
| Meta-MGNN | 82.94±0.10 | 82.13±0.13 | 75.43±0.21 | 73.36±0.32 | 68.99±1.84 | 65.54±2.13 | 76.27±0.57 | 72.43±0.85 | 72.58±0.34 | 72.51±0.35 |
| Pre-PAR | 84.93±0.11 | 83.01±0.09 | 78.08±0.16 | 74.46±0.29 | 69.96±1.37 | 66.94±1.12 | 75.12±0.84 | 73.63±1.00 | 73.71±0.61 | 72.49±0.61 |
| Pre-GS-Meta | 86.91±0.41 | 86.46±0.55 | 85.08±0.54 | 84.45±0.26 | 70.18±1.25 | 67.15±2.04 | 83.81±0.16 | 81.57±0.18 | 79.40±0.43 | 78.16±0.47 |
| Pre-ADKF-IFT | 86.06±0.35 | 80.97±0.48 | 70.95±0.60 | 62.16±1.03 | 95.74±0.37 | 67.25±3.87 | 76.22±0.13 | 71.13±1.15 | 80.21±0.27 | 76.62±0.73 |
| Pre-PACIA | 86.40±0.27 | 84.35±0.14 | 83.97±0.22 | 80.70±0.28 | 73.43±1.96 | 69.26±2.35 | 76.22±0.73 | 75.09±0.95 | 75.36±0.30 | 70.16±1.33 |
| Pre-KRGTS | 87.62±0.29 | 87.54±0.11 | 85.09±0.31 | 84.61±0.16 | 74.47±0.82 | 68.69±0.60 | 84.02±0.10 | 82.39±0.29 | 81.59±0.30 | 81.18±0.17 |
| Pin-Tuning | 91.56±2.57 | 85.71±1.33 | 93.41±3.52 | 84.36±1.73 | 73.33±2.00 | 66.44±2.50 | 84.94±1.09 | 80.08±1.21 | 81.26±0.46 | 71.98±1.55 |
| Uni-Match | 86.35±0.13 | – | 80.34±0.45 | – | 86.35±0.76 | – | 81.63±0.73 | – | – | – |
| **RECOG (Ours)** | **93.52±1.47** | **91.01±2.03** | **93.98±1.53** | **90.06±1.17** | **95.94±1.37** | **81.96±0.42** | **88.85±1.39** | **86.05±1.74** | **82.57±1.42** | **82.39±1.64** |

*Table 2.* Ablation study on the effects of different components evaluated on Tox21 and SIDER in terms of ROC-AUC (%).

| Method | Tox21 | | SIDER | |
|---|---|---|---|---|
| | 10-shot | 1-shot | 10-shot | 1-shot |
| w/o CPRL & CGIB | 90.21±2.31 | 86.91±1.49 | 90.19±3.89 | 86.93±1.42 |
| w/o CGIB | 91.46±3.55 | 90.28±2.68 | 91.54±3.74 | 88.42±1.73 |
| w/o CPRL | 91.00±0.42 | 88.18±1.04 | 91.60±1.26 | 87.57±0.86 |
| **RECOG (Ours)** | **93.52±1.47** | **91.01±2.03** | **93.98±1.53** | **90.06±1.17** |

## 5.2. Overall Performance

We present the performance comparison between our proposed RECOG and 16 baseline methods across five benchmark datasets under both 10-shot and 1-shot settings in Table 1. Based on the quantitative results, we have following observations: (1) the proposed RECOG consistently outperforms all baseline methods in all datasets and shot settings, while exhibiting lower variance in most cases. This indicates that RECOG not only improves average predictive performance but also yields stable results. (2) From the dataset perspective, the performance gains of RECOG are most pronounced on the MUV dataset. MUV is characterized by extremely sparsity, implying that both supervision and context information are strictly limited. The strong results on this dataset demonstrate that, under such information-scarce conditions, RECOG can effectively exploit the limited context information. (3) From the few-shot learning perspective, RECOG generally achieves larger relative improvements in the 1-shot setting compared to the 10-shot setting. In the 1-shot regime, the supervision signal is minimal. Although context learning can partially compensate for the lack of labeled data, this setting places higher demands on the quality of context signals, as redundant or noisy context may impair model performance. The superior performance of RECOG in this setting suggests that it can effectively utilize infor-

mative context while suppressing irrelevant or redundant context signals. Collectively, these results provide strong empirical evidence for the superiority of RECOG. Moreover, the above observations demonstrate that RECOG effectively addresses two key challenges faced by existing methods: *insufficient structural context modeling* and *redundant auxiliary context learning*.

## 5.3. Ablation Study

The proposed RECOG consists of two key components: the cross-property relation learning module (CPRL) and the context graph information bottleneck module (CGIB). To examine the contribution of each component, we conduct ablation studies by removing them individually. The results on the Tox21 and SIDER datasets are reported in Table 2. From the results, we observe that removing CPRL leads to the most significant performance degradation across both datasets, indicating that effectively exploiting context information is crucial for knowledge transfer and prediction accuracy improvement. This is reasonable because CPRL explicitly transforms cross-property contextual relations into auxiliary supervision signals, which provide additional learning cues beyond the limited target labels. Without CPRL, the model loses the ability to fully mine useful relational patterns from the context graph and therefore becomes more dependent on scarce task-specific supervision. In contrast, removing CGIB results in relatively smaller drops in average performance but noticeably increases performance variance, suggesting reduced training stability. This observation is consistent with the role of CGIB in filtering task-irrelevant context signals. Since auxiliary context inevitably contains noisy or weakly related information, directly using such context without bottleneck-based compression may introduce unstable supervision and disturb task-relevant representation

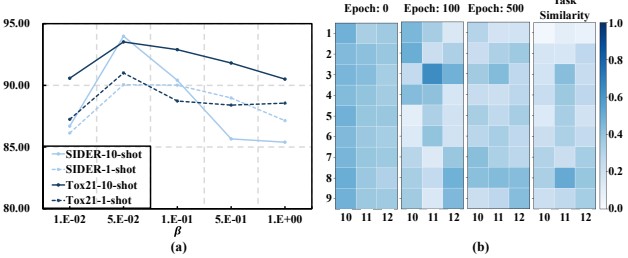

*Figure 4.* Analysis of context graph information bottleneck: (a) Effect of the information bottleneck coefficient $\beta$ on Tox21 and SIDER under the 10-shot and 1-shot settings; (b) The visualization of auxiliary task retain probability on Tox21 under 10-shot.

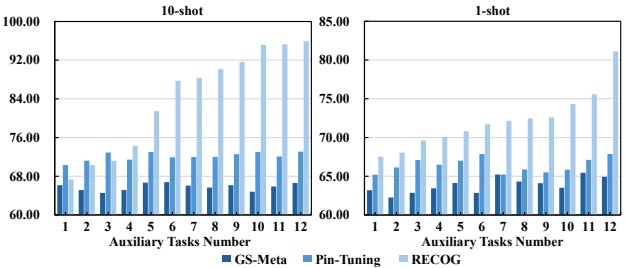

*Figure 5.* The performance on the MUV dataset with varying numbers of auxiliary tasks.

learning. Therefore, CGIB mainly contributes to improving the robustness and stability of the model, rather than directly increasing predictive accuracy. Overall, the ablation results demonstrate that CPRL and CGIB play complementary roles in ReCoG: CPRL enhances the utilization of informative cross-property context, while CGIB suppresses task-irrelevant auxiliary information. Their collaboration is essential for achieving optimal performance.

### 5.4. In-depth Analysis of ReCoG

**Hyper-parameter Sensitivity Analysis on $\beta$.** We further examine the influence of the hyperparameter $\beta$, which regulates the strength of the context graph information bottleneck in the overall optimization objective defined in Eq. (8). The value of $\beta$ is varied over $\{1.\text{E} - 02, 5.\text{E} - 02, 1.\text{E} - 01, 5.\text{E} - 01, 1.\text{E} + 00\}$, and the corresponding results on the Tox21 and SIDER datasets under the 1-shot setting are illustrated in Figure 4(a). As $\beta$ decreases, the bottleneck constraint becomes insufficient to suppress task-irrelevant context noise, leading to degraded predictive performance. In contrast, excessively large $\beta$ values enforce overly strong compression on the environment subgraph, which discards informative auxiliary signals and thus harms performance. The model achieves its best performance at $\beta = 5.\text{E} - 02$, where the trade-off between retaining useful context information and filtering redundant noise is well balanced.

*Table 3.* Effect of the Gumbel-Sigmoid temperature coefficient $t$.

| Dataset | Shot | 1.E − 01 | 2.E − 01 | 5.E − 01 | 1.E + 00 | 2.E + 00 |
|---------|------|----------|----------|----------|----------|----------|
| Tox21 | 10 | 89.37±1.67 | 90.00±1.56 | 91.25±1.50 | **93.52±1.47** | 91.47±1.23 |
| | 1 | 88.62±1.55 | 89.17±1.63 | 89.58±1.91 | **91.01±2.03** | 88.69±4.28 |
| SIDER | 10 | 89.40±3.07 | 92.62±4.39 | 92.12±0.24 | **93.98±1.53** | 90.13±1.05 |
| | 1 | 85.44±1.78 | 85.98±1.65 | 87.48±1.77 | **90.06±1.17** | 88.37±3.21 |

**Hyper-parameter Sensitivity Analysis on $t$.** We further investigate the influence of the Gumbel-Sigmoid temperature coefficient $t$, which controls the smoothness of the discrete retention probability estimation in the information bottleneck module. The $t$ is varied over $\{1.\text{E} - 01, 2.\text{E} - 01, 5.\text{E} - 01, 1.\text{E} + 00, 2.\text{E} + 00\}$, and the corresponding results on the Tox21 and SIDER datasets under both the 10-shot and 1-shot settings are reported in Table 3. As $t$ becomes excessively small, the gating distribution becomes overly discrete, resulting in unstable optimization and insufficient exploration of informative auxiliary-task signals. In contrast, overly large $t$ produce excessively smooth stochastic gates, weakening the bottleneck's ability to selectively filter task-irrelevant context information. The model consistently achieves the best performance at the $t = 1.\text{E} + 00$, suggesting that it provides a favorable trade-off between stable optimization and effective context-aware information filtering.

**Visualization of Context Graph Information Bottleneck.** To further investigate why ReCoG is effective, we visualize the retain probabilities produced by the context graph information bottleneck for auxiliary-task information across different target tasks during training. Figure 4 (b) reports results on Tox21 under the 10-shot setting. In particular, the fourth heatmap of Figure 4 (b) shows the label-distribution similarity among properties (cosine similarity), which serves as a reference for property relations. We observe that as training proceeds, the bottleneck decisions progressively align with the dataset's inherent task similarity, suggesting that the learned gating mechanism is not arbitrary but reflects meaningful cross-property relations. More concretely, auxiliary tasks that are more related to the target task tend to receive consistently higher retention probabilities, while less-related tasks are gradually suppressed. This indicates that the bottleneck learns a task-adaptive compression policy that preserves beneficial context signals and compresses irrelevant information, which in turn stabilizes context learning and contributes to improving model generalization.

**Effectiveness under Varying Auxiliary Task Numbers.** To further investigate the effect of auxiliary tasks, we compare the performance of different methods under varying numbers of auxiliary tasks on the MUV dataset. The results are shown in Figure 5. As the number of auxiliary tasks increases, the performance of ReCoG improves consistently under both the 10-shot and 1-shot settings, demonstrating its ability to effectively leverage additional auxiliary su-

*Table 4.* Performance comparison with different loss functions for relation learning.

| Method | Tox21 | | SIDER | |
|---|---|---|---|---|
| | 10-shot | 1-shot | 10-shot | 1-shot |
| Pre-GS-Meta | 86.91±0.41 | 86.46±0.55 | 85.08±0.54 | 84.45±0.26 |
| Pre-PACIA | 86.40±0.27 | 84.35±0.14 | 83.97±0.22 | 80.70±0.28 |
| Pre-KRGTS | 87.62±0.29 | 87.54±0.11 | 85.09±0.31 | 84.61±0.16 |
| Pin-Tuning | 91.56±2.57 | 85.71±1.33 | 93.41±3.52 | 84.36±1.73 |
| RECOG-BCE | 91.50±1.66 | 88.79±3.48 | 91.79±0.86 | 87.71±2.76 |
| **RECOG** | **93.52±1.47** | **91.01±2.03** | **93.98±1.53** | **90.06±1.17** |

*Table 5.* Training time comparison on TOX21, SIDER, and MUV in seconds with Tesla V100 GPU.

| Method | Tox21 | | SIDER | | MUV | |
|---|---|---|---|---|---|---|
| | 10-shot | 1-shot | 10-shot | 1-shot | 10-shot | 1-shot |
| Pre-GS-Meta | 3.13s (86.91) | 2.98s (86.46) | 2.90s (85.08) | 2.71s (84.45) | 3.27s (70.18) | 3.08s (67.15) |
| Pre-KRGTS | 4.28s (87.62) | 3.30s (87.54) | 6.31s (85.09) | 4.71s (84.61) | 5.83s (74.47) | 4.33s (68.69) |
| Pin-Tuning | 1.34s (91.56) | 1.08s (85.71) | 1.12s (93.41) | 1.06s (84.36) | 1.20s (73.33) | 1.13s (66.44) |
| **RECOG (Ours)** | 1.78s (93.52) | 1.65s (91.01) | 1.58s (93.98) | 1.38s (90.06) | 1.56s (95.94) | 1.37s (81.96) |

pervision. In contrast, the improvements of GS-Meta and Pin-Tuning remain relatively limited and unstable as more auxiliary tasks are introduced. More specifically, under the 10-shot setting, RECOG achieves substantial performance gains when increasing the number of auxiliary tasks from 1 to 12, whereas the competing methods exhibit only marginal improvements. A similar trend can also be observed in the more challenging 1-shot setting, where RECOG maintains a steady upward trend while the baselines benefit much less from additional auxiliary tasks. These results indicate that simply introducing more auxiliary tasks does not necessarily improve performance. Instead, the model must be capable of effectively exploiting useful inter-task knowledge while mitigating the interference caused by irrelevant auxiliary supervision. The superior scalability of RECOG with respect to the number of auxiliary tasks further demonstrates its effectiveness in modeling task relationships and selectively utilizing informative context knowledge.

**Analysis on the Loss Function Design of CPRL.** We further analyze the choice of the loss function used for the CPRL module in RECOG. Although the auxiliary supervision is derived from binary property labels, the objective of this component is not standard binary classification. Instead, it aims to model cross-property relation signals that reflect the degree of semantic consistency between molecular responses under different property dimensions. From this perspective, the learning target can be regarded as a relation regression signal, making the MSE loss a natural choice for optimizing such continuous relation-level supervision. To empirically validate this design, we compare RECOG with a BCE-based variant, denoted as RECOG-BCE, where the MSE loss is replaced by binary cross-entropy while keeping the remaining components unchanged. The results on Tox21 and SIDER datasets are reported in Table 4. We observe that RECOG-BCE still outperforms several strong baselines in most settings, indicating that the proposed relation learning

framework is effective regardless of the specific loss form. More importantly, RECOG consistently achieves better performance than RECOG-BCE across different datasets and shot settings. This demonstrates that MSE is more suitable for the relation modeling objective in RECOG, as it provides a smoother optimization signal for capturing cross-property relations rather than enforcing hard binary classification.

### 5.5. Running Time Comparison

We further compare the computational efficiency of RECOG with existing baselines in terms of training time, as reported in Table 5. Overall, RECOG achieves a favorable trade-off between predictive performance and computational cost, showing competitive training efficiency while outperforming most baselines. Although RECOG is slightly slower than Pin-Tuning, this is expected since Pin-Tuning adopts parameter-efficient adaptation with only a small number of trainable parameters, whereas RECOG additionally involves cross-property relation modeling and information bottleneck regularization. Importantly, this marginal increase in training cost brings substantial performance gains. Across all evaluated datasets and shot settings, RECOG achieves an average relative improvement of 11.19% in ROC-AUC compared with Pin-Tuning, with the largest gain of 25.74% observed on MUV. These results demonstrate that RECOG improves predictive performance significantly while keeping the additional computational overhead acceptable.

## 6. Conclusion

In this paper, we propose a novel framework, named RECOG, to learn on relational and compact context graphs for effective few-shot molecular property prediction performance. We first identify the critical challenges of insufficient exploitation and redundant information in existing context graph based methods, and then provide theoretical demonstration for the optimization objective of joint relational and compact knowledge extraction in context graphs. With the proposed cross-property relation learning module and the context graph information bottleneck module, our RECOG enhances the utilization of informative context relations and filtering task-irrelevant auxiliary signals, achieving consistently superior experiment results over multiple benchmarks. A potential limitation of our work lies in its relatively high variance on some datasets, which may be partly related to the stochastic episodic sampling strategy and the additional instability and computational overhead associated with bi-level optimization. In future work, we will explore more structured meta-learning schedules, deterministic context sampling strategies, and efficient gradient approximation methods to further stabilize training, reduce computational overhead, and improve robustness.

## Acknowledgements

Shanqing Yu is supported in part by the National Key R&D Program of China under Grant (2025YFA1510900), by the Baima Lake Laboratory Joint Fund of Zhejiang Provincial Natural Science Foundation of China (LBMHZ25F020002), by the Yangtze River Delta Science and Technology Innovation Community Joint Research Project (2026ZY03003), by the Key Research and Development Program of Zhejiang Province (2024C01025). Qi Xuan is supported in part by the Yangtze River Delta Science and Technology Innovation Community Joint Research Project (2025CSJGG01000).

## Impact Statement

This paper presents work whose goal is to advance the field of Machine Learning, with a focus on few-shot molecular property prediction. By improving data-efficient molecular modeling, this work may indirectly support research related to drug discovery and other life and health sciences, potentially contributing to societal benefits in healthcare and biomedical research. These impacts are contingent on responsible use and experimental validation. We do not identify any specific ethical concerns or broader consequences that require further discussion.

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

## A. Proofs

### A.1. Proofs of Equation (11)

By the definition of conditional mutual information, we have:

$$
\begin{aligned}
I(\mathcal{G}_\tau; Y_{\mathrm{rel}} \mid Y_\tau) &= H(Y_{\mathrm{rel}} \mid Y_\tau) - H(Y_{\mathrm{rel}} \mid \mathcal{G}_\tau, Y_\tau) \\
&= H(Y_{\mathrm{rel}} \mid Y_\tau) + \mathbb{E}_{\mathcal{G}_\tau, Y_\tau, Y_{\mathrm{rel}}}\Big[\log p(Y_{\mathrm{rel}} \mid \mathcal{G}_\tau, Y_\tau)\Big]
\end{aligned}
\tag{28}
$$

Directly optimizing $\log p(Y_{\mathrm{rel}} \mid \mathcal{G}_\tau, Y_\tau)$ is generally intractable. By introducing a variational distribution $q_\phi(Y_{\mathrm{rel}} \mid \mathcal{G}_\tau)$ to approximate the intractable true posterior $p(Y_{\mathrm{rel}} \mid \mathcal{G}_\tau, Y_\tau)$, we obtain a valid variational lower bound by the non-negativity of the KL divergence. Then, we can rewrite the negative conditional entropy term as follows:

$$
\begin{aligned}
-H(Y_{\mathrm{rel}} \mid \mathcal{G}_\tau, Y_\tau) &= \mathbb{E}_{p(\mathcal{G}_\tau, Y_\tau, Y_{\mathrm{rel}})}[\log p(Y_{\mathrm{rel}} \mid \mathcal{G}_\tau, Y_\tau)] \\
&= \mathbb{E}_{p(\mathcal{G}_\tau, Y_\tau, Y_{\mathrm{rel}})}\left[\log\left(\frac{p(Y_{\mathrm{rel}} \mid \mathcal{G}_\tau, Y_\tau)}{q_\phi(Y_{\mathrm{rel}} \mid \mathcal{G}_\tau)} \cdot q_\phi(Y_{\mathrm{rel}} \mid \mathcal{G}_\tau)\right)\right] \\
&= \mathbb{E}_{p(\mathcal{G}_\tau, Y_\tau)}[D_{KL}(p(Y_{\mathrm{rel}} \mid \mathcal{G}_\tau, Y_\tau)\|q_\phi(Y_{\mathrm{rel}} \mid \mathcal{G}_\tau))] + \mathbb{E}_{p(\mathcal{G}_\tau, Y_\tau, Y_{\mathrm{rel}})}[\log q_\phi(Y_{\mathrm{rel}} \mid \mathcal{G}_\tau)].
\end{aligned}
$$

According to the non-negativity of the KL divergence, i.e., $D_{KL}(\cdot\|\cdot) \geq 0$, we have the following inequality:

$$
-H(Y_{\mathrm{rel}} \mid \mathcal{G}_\tau, Y_\tau) \geq \mathbb{E}_{\mathcal{G}_\tau, Y_\tau, Y_{\mathrm{rel}}}[\log q_\phi(Y_{\mathrm{rel}} \mid \mathcal{G}_\tau)].
\tag{29}
$$

By plugging Equation 29 into Equation 28, we obtain the final variational lower bound:

$$
I(\mathcal{G}_\tau; Y_{\mathrm{rel}} \mid Y_\tau) \geq \mathbb{E}_{\mathcal{G}_\tau, Y_\tau, Y_{\mathrm{rel}}}[\log q_\phi(Y_{\mathrm{rel}} \mid \mathcal{G}_\tau)] + H(Y_{\mathrm{rel}} \mid Y_\tau).
\tag{30}
$$

### A.2. Proofs of Equation (18)

By the definition of mutual information, we have

$$
\begin{aligned}
I\left(\mathcal{G}_\tau^{\mathrm{task}} \cup \widetilde{\mathcal{G}}_\tau^{\mathrm{env}}; Y_\tau\right) &= H(Y_\tau) - H\left(Y_\tau \mid \mathcal{G}_\tau^{\mathrm{task}} \cup \widetilde{\mathcal{G}}_\tau^{\mathrm{env}}\right) \\
&= H(Y_\tau) + \mathbb{E}_{p(\mathcal{G}_\tau^{\mathrm{task}}, \widetilde{\mathcal{G}}_\tau^{\mathrm{env}}, Y_\tau)}\left[\log p\left(Y_\tau \mid \mathcal{G}_\tau^{\mathrm{task}} \cup \widetilde{\mathcal{G}}_\tau^{\mathrm{env}}\right)\right].
\end{aligned}
\tag{31}
$$

Since the true conditional distribution $p\left(Y_\tau \mid \mathcal{G}_\tau^{\mathrm{task}} \cup \widetilde{\mathcal{G}}_\tau^{\mathrm{env}}\right)$ is generally intractable, we introduce a variational distribution $q_\theta\left(Y_\tau \mid \mathcal{G}_\tau^{\mathrm{task}} \cup \widetilde{\mathcal{G}}_\tau^{\mathrm{env}}\right)$ to approximate it. Then,

$$
\begin{aligned}
&\mathbb{E}_{p(\mathcal{G}_\tau^{\mathrm{task}}, \widetilde{\mathcal{G}}_\tau^{\mathrm{env}}, Y_\tau)}\left[\log p\left(Y_\tau \mid \mathcal{G}_\tau^{\mathrm{task}} \cup \widetilde{\mathcal{G}}_\tau^{\mathrm{env}}\right)\right] \\
&= \mathbb{E}_{p(\mathcal{G}_\tau^{\mathrm{task}}, \widetilde{\mathcal{G}}_\tau^{\mathrm{env}}, Y_\tau)}\left[\log \frac{p\left(Y_\tau \mid \mathcal{G}_\tau^{\mathrm{task}} \cup \widetilde{\mathcal{G}}_\tau^{\mathrm{env}}\right)}{q_\theta\left(Y_\tau \mid \mathcal{G}_\tau^{\mathrm{task}} \cup \widetilde{\mathcal{G}}_\tau^{\mathrm{env}}\right)} + \log q_\theta\left(Y_\tau \mid \mathcal{G}_\tau^{\mathrm{task}} \cup \widetilde{\mathcal{G}}_\tau^{\mathrm{env}}\right)\right] \\
&= \mathbb{E}_{p(\mathcal{G}_\tau^{\mathrm{task}}, \widetilde{\mathcal{G}}_\tau^{\mathrm{env}})}\left[D_{\mathrm{KL}}\left(p\left(Y_\tau \mid \mathcal{G}_\tau^{\mathrm{task}} \cup \widetilde{\mathcal{G}}_\tau^{\mathrm{env}}\right) \Big\| q_\theta\left(Y_\tau \mid \mathcal{G}_\tau^{\mathrm{task}} \cup \widetilde{\mathcal{G}}_\tau^{\mathrm{env}}\right)\right)\right] \\
&\quad + \mathbb{E}_{p(\mathcal{G}_\tau^{\mathrm{task}}, \widetilde{\mathcal{G}}_\tau^{\mathrm{env}}, Y_\tau)}\left[\log q_\theta\left(Y_\tau \mid \mathcal{G}_\tau^{\mathrm{task}} \cup \widetilde{\mathcal{G}}_\tau^{\mathrm{env}}\right)\right] \\
&\geq \mathbb{E}_{p(\mathcal{G}_\tau^{\mathrm{task}}, \widetilde{\mathcal{G}}_\tau^{\mathrm{env}}, Y_\tau)}\left[\log q_\theta\left(Y_\tau \mid \mathcal{G}_\tau^{\mathrm{task}} \cup \widetilde{\mathcal{G}}_\tau^{\mathrm{env}}\right)\right],
\end{aligned}
\tag{32}
$$

where the inequality follows from the non-negativity of KL divergence, $D_{\mathrm{KL}}(\cdot\|\cdot) \geq 0$. Finally, substituting Eq. (32) into Eq. (31) yields the variational lower bound:

$$
I\left(\mathcal{G}_\tau^{\mathrm{task}} \cup \widetilde{\mathcal{G}}_\tau^{\mathrm{env}}; Y_\tau\right) \geq \mathbb{E}_{p(\mathcal{G}_\tau^{\mathrm{task}}, \widetilde{\mathcal{G}}_\tau^{\mathrm{env}}, Y_\tau)}\left[\log q_\theta\left(Y_\tau \mid \mathcal{G}_\tau^{\mathrm{task}} \cup \widetilde{\mathcal{G}}_\tau^{\mathrm{env}}\right)\right] + H(Y_\tau).
\tag{33}
$$

### A.3. Proofs of Equation (23)

Let $\widetilde{\mathcal{G}}_\tau^{\mathrm{env}}$ be the perturbed context subgraph, and $\mathcal{G}_\tau^{\mathrm{env}}, \mathcal{G}_\tau^{\mathrm{task}}$ be the original graphs. Let $z_{\widetilde{\mathcal{G}}_\tau^{\mathrm{env}}} \in \mathbb{R}^{d_1}$ denote the readout representation of $\widetilde{\mathcal{G}}_\tau^{\mathrm{env}}$. Following the common practice in VIB (Alemi et al., 2017), we assume the readout process is information-preserving, i.e., $I(\widetilde{\mathcal{G}}_\tau^{\mathrm{env}}; \mathcal{G}_\tau^{\mathrm{env}}, \mathcal{G}_\tau^{\mathrm{task}}) \approx I(z_{\widetilde{\mathcal{G}}_\tau^{\mathrm{env}}}; \mathcal{G}_\tau^{\mathrm{env}}, \mathcal{G}_\tau^{\mathrm{task}})$. By the definition of mutual information and the variational approximation $q(z_{\widetilde{\mathcal{G}}_\tau^{\mathrm{env}}})$, the following identity holds:

$$
\begin{aligned}
I(z_{\widetilde{\mathcal{G}}_\tau^{\mathrm{env}}}; \mathcal{G}_\tau^{\mathrm{env}}, \mathcal{G}_\tau^{\mathrm{task}}) &= \mathbb{E}_{\mathcal{G}_\tau^{\mathrm{env}}, \mathcal{G}_\tau^{\mathrm{task}}}\Big[ D_{\mathrm{KL}}\big(p_\phi(z_{\widetilde{\mathcal{G}}_\tau^{\mathrm{env}}} \mid \mathcal{G}_\tau^{\mathrm{env}}, \mathcal{G}_\tau^{\mathrm{task}})\|p(z_{\widetilde{\mathcal{G}}_\tau^{\mathrm{env}}})\big) \Big] \\
&= \mathbb{E}_{\mathcal{G}_\tau^{\mathrm{env}}, \mathcal{G}_\tau^{\mathrm{task}}}\Big[ D_{\mathrm{KL}}\big(p_\phi(z_{\widetilde{\mathcal{G}}_\tau^{\mathrm{env}}} \mid \mathcal{G}_\tau^{\mathrm{env}}, \mathcal{G}_\tau^{\mathrm{task}})\|q(z_{\widetilde{\mathcal{G}}_\tau^{\mathrm{env}}})\big) - D_{\mathrm{KL}}\big(p(z_{\widetilde{\mathcal{G}}_\tau^{\mathrm{env}}})\|q(z_{\widetilde{\mathcal{G}}_\tau^{\mathrm{env}}})\big) \Big] \\
&\le \mathbb{E}_{\mathcal{G}_\tau^{\mathrm{env}}, \mathcal{G}_\tau^{\mathrm{task}}}\Big[ D_{\mathrm{KL}}\big(p_\phi(z_{\widetilde{\mathcal{G}}_\tau^{\mathrm{env}}} \mid \mathcal{G}_\tau^{\mathrm{env}}, \mathcal{G}_\tau^{\mathrm{task}})\|q(z_{\widetilde{\mathcal{G}}_\tau^{\mathrm{env}}})\big) \Big].
\end{aligned}
\tag{34}
$$

Given that $X^{\mathrm{env}}$ denotes the embeddings of $\mathcal{G}_\tau^{\mathrm{env}}$ and the perturbed node features $\widetilde{\boldsymbol{X}}_{\tau'}^{\mathrm{env}} = \lambda_{\tau'}\boldsymbol{X}_{\tau'}^{\mathrm{env}} + (1 - \lambda_{\tau'})\epsilon, (\epsilon \sim \mathcal{N}(\mu_{\boldsymbol{X}^{\mathrm{env}}}, \sigma_{\boldsymbol{X}^{\mathrm{env}}}^2))$, we have:

$$
q(z_{\widetilde{\mathcal{G}}_\tau^{\mathrm{env}}}) = \mathcal{N}\Big(N_{\mathrm{auxi}}\mu_{\boldsymbol{X}^{\mathrm{env}}}, N_{\mathrm{auxi}}\sigma_{\boldsymbol{X}^{\mathrm{env}}}^2\Big).
\tag{35}
$$

Similarly,

$$
p_\phi(z_{\widetilde{\mathcal{G}}_\tau^{\mathrm{env}}} \mid \mathcal{G}_\tau^{\mathrm{env}}, \mathcal{G}_\tau^{\mathrm{task}}) = \mathcal{N}\Big(N_{\mathrm{auxi}}\mu_{\boldsymbol{X}^{\mathrm{env}}} + \sum_{\tau'=1}^{N_{\mathrm{auxi}}}\lambda_{\tau'}(X_\tau'^{\mathrm{env}} - \mu_{\boldsymbol{X}^{\mathrm{env}}}), \sum_{\tau'=1}^{N_{\mathrm{auxi}}}(1 - \lambda_{\tau'})^2\sigma_{\boldsymbol{X}^{\mathrm{env}}}^2\Big).
\tag{36}
$$

Using the KL divergence between two Gaussians, we can derive the KL divergence as:

$$
D_{\mathrm{KL}}\Big(p_\phi(\cdot \mid \mathcal{G}_\tau^{\mathrm{env}}, \mathcal{G}_\tau^{\mathrm{task}}) \,\Big\|\, q(\cdot)\Big) = -\frac{d_1}{2}\log A + \frac{d_1}{2N_{\mathrm{auxi}}}A + \frac{1}{2N_{\mathrm{auxi}}}\|B\|_2^2 + C,
\tag{37}
$$

where $A = \sum_{\tau'=1}^{N_{\mathrm{auxi}}}(1 - \lambda_{\tau'})^2, B = \frac{\sum_{\tau'=1}^{N_{\mathrm{auxi}}}\lambda_{\tau'}(X_{\tau'}^{\mathrm{env}} - \mu_{\boldsymbol{X}^{\mathrm{env}}})}{\sigma_{\boldsymbol{X}^{\mathrm{env}}}}$, $C$ is a constant term. Plugging Eq. (37) into Eq. (34) yields:

$$
\begin{aligned}
I(\widetilde{\mathcal{G}}_\tau^{\mathrm{env}}; \mathcal{G}_\tau^{\mathrm{env}}, \mathcal{G}_\tau^{\mathrm{task}}) &\le \mathbb{E}_{\mathcal{G}_\tau^{\mathrm{env}}, \mathcal{G}_\tau^{\mathrm{task}}}\left[ -\frac{d_1}{2}\log A + \frac{d_1}{2N_{\mathrm{auxi}}}A + \frac{1}{2N_{\mathrm{auxi}}}\|B\|_2^2 \right] \\
&:= \mathcal{L}_{\mathrm{MI}}
\end{aligned}
\tag{38}
$$

## B. Algorithm

To facilitate a clearer understanding of the training procedure, we summarize it in concise pseudo-code in Algorithm 1.

---

**Algorithm 1** The algorithm of RECOG.

---

**Require:** Training set $\mathcal{D}_{\mathrm{train}}$
**Ensure:** Tuned few-shot molecular property prediction model with parameter $\zeta$
 1: **while** not converge **do**
 2:     Sample $B$ episodes from training set $\mathcal{D}_{\mathrm{train}}$ to form a mini-batch $\{E_\tau\}_{\tau=1}^B$;
 3:     **for** $\tau = 1$ to $B$ **do**
 4:         Calculate classification loss on support set $\mathcal{L}_{\mathrm{pred}}^{\mathcal{S}_\tau}$ by Equation (24) on $E_\tau$;
 5:         Do inner-loop update by Equation (25) on $E_\tau$ by Equation (25);
 6:         Calculate classification loss on query set $\mathcal{L}_{\mathrm{total}}^{Q_\tau}$ by Equation (26) on $E_\tau$;
 7:     **end for**
 8:     Do outer-loop optimization by Equation (25);
 9: **end while**
10: **Return** optimized model parameter $\zeta$.

---

*Table 6.* Statistics of the five MoleculeNet datasets used for few-shot molecular property prediction, including the number of compounds, tasks, task splits, and label distributions.

| Dataset | Tox21 | SIDER | MUV | ToxCast | PCBA |
|---|---|---|---|---|---|
| #Compound | 7,831 | 1,427 | 9,312 | 8,575 | 437,929 |
| #Property | 12 | 27 | 17 | 617 | 128 |
| #Train Property | 9 | 21 | 12 | 451 | 118 |
| #Test Property | 3 | 6 | 5 | 158 | 10 |
| %Label Active | 6.24 | 56.76 | 0.31 | 12.60 | 0.84 |
| %Label Inactive | 76.71 | 43.24 | 15.76 | 72.43 | 59.84 |
| %Unknown Label | 17.05 | 0 | 84.21 | 14.97 | 39.32 |

*Table 7.* Statistics of ToxCast sub-datasets grouped by assay provider

| | APR | ATG | BSK | CEETOX | CLD | NVS | OT | TOX21 | Tanguay |
|---|---|---|---|---|---|---|---|---|---|
| #Compound | 1,039 | 3,423 | 1,445 | 508 | 305 | 2,130 | 1,782 | 8,241 | 1,039 |
| #Property | 43 | 146 | 115 | 14 | 19 | 139 | 15 | 100 | 18 |
| #Train Property | 33 | 106 | 84 | 10 | 14 | 100 | 11 | 80 | 13 |
| #Test Property | 10 | 40 | 31 | 4 | 5 | 39 | 4 | 20 | 5 |
| %Label Active | 10.30 | 5.92 | 17.71 | 22.26 | 30.72 | 3.21 | 9.78 | 5.39 | 8.05 |
| %Label Inactive | 61.61 | 93.92 | 82.29 | 76.38 | 68.30 | 94.52 | 87.78 | 86.26 | 90.84 |
| %Missing Label | 28.09 | 0.16 | 0.00 | 1.36 | 0.98 | 2.27 | 2.44 | 8.35 | 1.11 |

# C. Experiment Settings

## C.1. Details of Datasets

We conduct experiments on five benchmark datasets from MoleculeNet (Wu et al., 2018), which are widely used for few-shot molecular property prediction. All datasets consist of multiple binary classification tasks, where each task corresponds to a distinct molecular property:

- **Tox21:** This dataset comprises 12 toxicity-related tasks derived from in vitro assays, aiming to predict potential toxic effects of chemical compounds.

- **SIDER:** This dataset includes 27 side-effect prediction tasks collected from marketed drugs, focusing on adverse drug reactions.

- **MUV:** This is a challenging benchmark consisting of 17 tasks with highly imbalanced labels, specifically designed to reduce dataset bias and false positives.

- **ToxCast:** This dataset consists of a large collection of toxicity screening tasks obtained from high-throughput assays, covering a wide range of biological targets.

- **PCBA:** This dataset contains numerous bioassay-based tasks for biological activity prediction, exhibiting substantial task and data diversity.

Following prior FSMPP studies (Zhuang et al., 2023; Wang et al., 2024a; 2025a), each dataset is formulated under a task-based learning setting. Tasks are split into meta-training and meta-testing sets with no overlap in molecular properties. During episodic training, 2-way $K$-shot classification tasks are constructed by sampling support and query sets from each task. Dataset statistics and detailed split settings are summarized in Table 6 and Table 7.

## C.2. Details of Baselines

We provide additional details of the baseline methods used for comparison, which can be broadly categorized into two paradigms based on whether pretrained molecular encoders are employed.

**Methods trained from scratch.** This group includes representative meta-learning and metric-learning approaches that learn molecular representations solely from the few-shot training data:

*Table 8.* 10-shot performance on each sub-dataset of ToxCast in terms of ROC-AUC (%).

| Model | APR | ATG | BSK | CEETOX | CLD | NVS | OT | TOX21 | Tanguay |
|---|---|---|---|---|---|---|---|---|---|
| ProtoNet | 73.58 | 59.26 | 70.15 | 66.12 | 78.12 | 65.85 | 64.90 | 68.26 | 73.61 |
| MAML | 72.66 | 62.09 | 66.42 | 64.08 | 74.57 | 66.56 | 64.07 | 68.04 | 77.12 |
| EGNN | 80.33 | 66.17 | 73.43 | 66.51 | 78.85 | 71.05 | 68.21 | 76.40 | 85.23 |
| Pre-PAR | 86.09 | 72.72 | 82.45 | 72.12 | 83.43 | 74.94 | 71.96 | 82.81 | 88.20 |
| Pre-GS-Meta | 90.15 | 82.54 | 88.21 | 74.19 | 86.34 | 76.29 | 74.47 | 90.63 | 91.47 |
| Pre-KRGTS | 90.31 | 80.12 | 87.92 | 76.63 | 86.97 | **77.52** | 75.11 | 89.83 | 91.73 |
| Pin-Tuning | 92.78 | 83.58 | 89.49 | 75.96 | 87.70 | 76.33 | 75.56 | 90.80 | 92.25 |
| RGCL(Ours) | **92.97** | **84.37** | **89.95** | **85.85** | **92.63** | 77.19 | **87.50** | **92.35** | **96.83** |

- **Siamese Network** (Koch et al., 2015) learns pairwise similarity functions for classification.

- **ProtoNet** (Snell et al., 2017) performs classification based on distances to class prototypes in the embedding space.

- **MAML** (Finn et al., 2017) optimizes model parameters for fast adaptation to new tasks via gradient-based optimization.

- **TPN** (Liu et al., 2019) exploits transductive inference by modeling task-level relationships among query samples.

- **EGNN** (Kim et al., 2019) incorporates graph neural networks to encode molecular structures in an end-to-end manner.

- **IterRefLSTM** (Altae-Tran et al., 2017) iteratively refines molecular representations using recurrent architectures.

- **MHNfs** (Schimunek et al., 2023) employs memory-based hypernetworks to enhance model generalization across tasks.

**Methods with pretrained molecular encoders.** The second group leverages molecular encoders pretrained on large-scale chemical datasets, followed by task-specific adaptation:

- **Pre-GNN** (Hu et al., 2020) utilizes GNNs pretrained via self-supervised objectives on molecular graphs.

- **Meta-MGNN** (Guo et al., 2021) combines pretrained graph encoders with meta-learning strategies for FSMPP.

- **Pre-PAR** (Wang et al., 2021) models molecular relations to improve transferability across molecular properties.

- **Pre-GS-Meta** (Zhuang et al., 2023) integrates context-aware meta-learning with pretrained representations.

- **Pre-ADKF-IFT** (Chen et al., 2023) adopts an adaptive kernel fusion mechanism for task-level knowledge transfer.

- **Pre-PACIA** (Wu et al., 2024) introduces context-aware interaction modeling to enhance cross-task generalization.

- **Pin-Tuning** (Wang et al., 2024a) performs parameter-efficient tuning by optimizing a small set of task-specific adapters.

- **Pre-KRGTS** (Wang et al., 2025a) explicitly captures task–task relations through a knowledge-enhanced relation graph.

- **Uni-Match** (Li et al., 2025a) proposed a context-aware matching network FSMPP methods.

For all baseline methods, we primarily report the results as presented in their original papers under the standard FSMPP settings. When official implementations are available, we directly adopt the reported results; otherwise, we rely on the experimental protocols and configurations described in the corresponding publications to ensure fair comparison.

### C.3. Details of Implementation

**Experimental environment.** All experiments are conducted on Linux-based servers equipped with Intel(R) Xeon(R) Gold 6140 CPUs (2.30 GHz), 377 GB RAM, and 8 Tesla V100 PCIe GPUs with 32 GB memory each. Our implementation is based on PyTorch 2.7.1 and PyTorch Geometric 2.6.1, running with CUDA 11.8. Molecular preprocessing and feature extraction are performed using RDKit 2025.9.2, and all experiments are executed under Python 3.9.25.

*Table 9.* 1-shot performance on each sub-dataset of ToxCast in terms of ROC-AUC (%).

| Method | APR | ATG | BSK | CEETOX | CLD | NVS | OT | TOX21 | Tanguay |
|---|---|---|---|---|---|---|---|---|---|
| ProtoNet | 57.08 | 54.92 | 53.92 | 60.25 | 66.25 | 54.87 | 63.11 | 58.27 | 58.32 |
| MAML | 64.59 | 55.45 | 60.36 | 61.02 | 66.22 | 59.84 | 62.15 | 59.52 | 60.92 |
| EGNN | 67.06 | 57.28 | 60.82 | 60.10 | 71.53 | 56.56 | 66.08 | 63.32 | 74.80 |
| Pre-PAR | 84.69 | 70.38 | 79.89 | 66.57 | 77.83 | 72.51 | 70.41 | 80.33 | 86.64 |
| Pre-GS-Meta | 89.49 | 81.69 | 87.28 | 68.55 | 78.69 | 74.36 | 73.56 | 89.46 | 91.10 |
| Pre-KRGTS | 89.45 | 79.54 | 86.84 | 72.50 | 81.21 | **76.63** | 73.68 | 89.51 | 92.15 |
| Pin-Tuning | 87.08 | 80.67 | 83.12 | 71.91 | 79.15 | 71.29 | 73.17 | 84.27 | 90.08 |
| RGCL(Ours) | **94.39** | **84.92** | **89.23** | **76.71** | **87.79** | 76.29 | **77.89** | **91.79** | **95.40** |

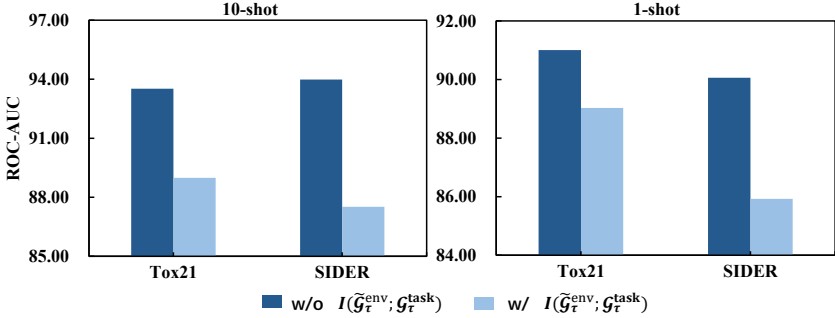

*Figure 6.* Effect of the mutual information term $I(\widetilde{\mathcal{G}}_\tau^{env}; \mathcal{G}_\tau^{task})$ on model performance. We observe that eliminating this term (w/o) consistently improves performance compared to including it (w/).

**Model Configuration.** Following prior work (Zhuang et al., 2023; Wang et al., 2024a; 2025a), we adopt standard molecular featurization schemes. Specifically, atomic number and chirality tags are used as atom features, while bond type and bond direction are employed as edge features, consistent with common practices in molecular pretraining. We use the pretrained GIN model released by Pre-GNN (Hu et al., 2020) as the molecular encoder, with the embedding dimension set to $d_1 = 300$, the layer number set to 5. The CONTEXTENCODER($\cdot$) described in Section 4.2 is implemented as a two-layer message passing neural network (MPNN) (Gilmer et al., 2017), with hidden dimension $d_2 = 300$. Within each MPNN layer, messages from neighboring nodes are aggregated through a linear transformation, and distinct edge features are incorporated to differentiate various edge types in the context graphs. The property classifier in Equation (10), the relation predictor in Equation (16), and the MLP (in Equation (20)) used in the context graph information bottleneck module are all implemented as two-layer MLPs with ReLU activation.

**Hyperparameter settings.** RECOG is trained for 2000 epochs using an episodic training scheme with a batch size of $B = 5$ episodes. The coefficient of the mutual information regularization term $\mathcal{L}_{MI}$ is fixed to $\beta = 5.E-02$. For meta-learning optimization, we tune both inner-loop and outer-loop learning rates. Specifically, the inner-loop learning rate is selected from $\{1.E-02, 5.E-02, 1.E-01, 5.E-01, 1.E+00\}$, while the outer-loop learning rate is searched over $\{5.E-05, 1.E-04, 5.E-04, 1.E-03, 5.E-03, 1.E-02, 5.E-02, 1.E-01\}$. In addition, the temperature coefficient $t$ used in the sigmoid function is tuned from $\{1.E-01, 2.E-01, 5.E-01, 1.E+00\}$.

## D. Additional Experiments

### D.1. The Impact of Maximizing $I(\widetilde{\mathcal{G}}_\tau^{env}; \mathcal{G}_\tau^{task})$

We investigate the effect of maximizing the mutual information term $I(\widetilde{\mathcal{G}}_\tau^{env}; \mathcal{G}_\tau^{task})$. While this is a natural objective for aligning the representations of $\widetilde{\mathcal{G}}_\tau^{env}$ and $\mathcal{G}_\tau^{task}$. Following prior work, we employ a contrastive learning mechanism to approximate it:

$$\mathcal{L}_{cont} = -\frac{1}{B} \sum_{\tau \in B} \log \frac{\exp(\text{sim}(z_{\widetilde{\mathcal{G}}_\tau^{env}}, z_{\mathcal{G}_\tau^{task}})/t)}{\sum_{\tau' \in B \setminus \tau} \exp(\text{sim}(z_{\widetilde{\mathcal{G}}_{\tau'}^{env}}, z_{\mathcal{G}_{\tau'}^{task}})/t)} \tag{39}$$

*Table 10.* Performance comparison on more challenging 1-shot datasets from Meta-MolNet.

| Method | JNK3 | HIV |
|---|---|---|
| Meta-GAT | 86.45±1.63 | 87.41±0.15 |
| ReCoG | **92.44±4.28** | **92.59±0.91** |

*Table 11.* Comparison with additional few-shot learning baselines.

| Method | Tox21 | | SIDER | |
|---|---|---|---|---|
| | 10-shot | 1-shot | 10-shot | 1-shot |
| MLTI | 78.95±0.17 | 65.49±2.45 | 81.57±1.72 | 65.85±1.48 |
| SMILE | 76.23±0.69 | 60.69±1.24 | 71.80±0.79 | 56.49±1.49 |
| ReCoG | **93.52±1.47** | **91.01±2.03** | **93.98±1.53** | **90.06±1.17** |

where $z_{\widetilde{\mathcal{G}}_\tau^{\mathrm{env}}}, z_{\mathcal{G}_\tau^{\mathrm{task}}}$ are the subgraph level representations of $\widetilde{\mathcal{G}}_\tau^{\mathrm{env}}, \mathcal{G}_\tau^{\mathrm{task}}$, respectively, $\mathrm{sim}(\cdot)$ denotes cosine similarity and $t$ is a temperature hyperparameter. Consequently, the outer-loop optimization can be reformulated as:

$$\zeta \leftarrow \zeta - \alpha_{\mathrm{outer}} \nabla_\zeta \Big( \frac{1}{B} \sum_{\tau=1}^{B} \mathcal{L}_{\mathrm{total}}^{Q_\tau} + \beta \mathcal{L}_{\mathrm{cont}} \Big), \tag{40}$$

Figure 6 reports the results on the Tox21 and SIDER datasets under both 10-shot and 1-shot settings, where "w/o $I(\widetilde{\mathcal{G}}_\tau^{\mathrm{env}}; \mathcal{G}_\tau^{\mathrm{task}})$" denotes our proposed ReCoG, while "w/ $I(\widetilde{\mathcal{G}}_\tau^{\mathrm{env}}; \mathcal{G}_\tau^{\mathrm{task}})$" represents the variant incorporating this additional optimization term. We observe a consistent and notable degradation in performance across all evaluated settings, indicating that explicitly enforcing such alignment is detrimental in our framework. We attribute this phenomenon to the *complementary* design of the context graph $\mathcal{G}_\tau$. Specifically, $\mathcal{G}_\tau^{\mathrm{task}}$ is designed to capture essential task-discriminative structure, whereas $\mathcal{G}_\tau^{\mathrm{env}}$ encodes auxiliary context patterns that are informative but should remain diverse. Forcibly increasing their mutual information causes the auxiliary semantics to redundantly align with the essential semantics, thereby reducing context diversity and undermining the model's ability to preserve heterogeneous context information. As a result, we omit this term and instead focus on minimizing $I(\widetilde{\mathcal{G}}_\tau^{\mathrm{env}}; \mathcal{G}_\tau^{\mathrm{env}}, \mathcal{G}_\tau^{\mathrm{task}})$, which provides a more appropriate regularization for filtering task-irrelevant context without inducing redundant alignment.

### D.2. Performance on Sub-datasets of ToxCast

The detailed comparisons between ReCoG and baseline methods on the ToxCast sub-datasets are summarized in Table 8 and Table 9. Results show that ReCoG achieves competitive or superior performance across ToxCast sub-datasets under both 10-shot and 1-shot settings, demonstrating stable generalization compared to baseline methods.

### D.3. Performance on More Challenging 1-shot Datasets

To further evaluate the generalization ability of ReCoG in more challenging few-shot scenarios, we conduct additional experiments on two difficult 1-shot datasets from Meta-MolNet (Lv et al., 2025), namely JNK3 and HIV. These datasets are commonly used for evaluating molecular few-shot learning methods and remain challenging due to extremely limited supervision and complex molecular-property relationships. We compare ReCoG with Meta-GAT, a strong baseline reported in Meta-MolNet, and summarize the results in Table 10. As shown in Table 10, ReCoG consistently outperforms Meta-GAT on both JNK3 and HIV, further demonstrating its generalization ability on more difficult 1-shot tasks.

### D.4. Comparison with Additional Few-shot Baselines

We further compare ReCoG with two additional few-shot baselines, MLTI (Yao et al., 2022) and SMILE (Liu et al., 2025), which improve generalization mainly through mixup-based data augmentation or task interpolation. As reported in Table 11, ReCoG consistently outperforms both methods on Tox21 and SIDER under different settings, with more pronounced gains in the more challenging 1-shot scenario. One possible reason is that direct sample interpolation or synthetic sample generation may be less suitable for molecular property prediction, as it can introduce chemically implausible samples, thereby weakening the reliability of few-shot supervision. In contrast, ReCoG exploits cross-property contextual relations without explicitly generating additional molecular samples, leading to more stable and effective knowledge transfer.

*Table 12.* $K$-shot sensitivity analysis on SIDER and Tox21.

| Method | SIDER | | | Tox21 | | |
|---|---|---|---|---|---|---|
| | 1-shot | 5-shot | 10-shot | 1-shot | 5-shot | 10-shot |
| Pin-Tuning | 84.36±1.73 | 92.02±3.01 | 93.41±3.52 | 85.71±1.33 | 90.95±2.33 | 91.56±2.57 |
| RECOG | **90.06±1.17** | **92.87±0.78** | **93.98±1.53** | **91.01±2.03** | **91.69±0.64** | **93.52±1.47** |

*Table 13.* PR-AUC comparison on Tox21 and SIDER under 10-shot and 1-shot settings.

| Method | Tox21 | | SIDER | |
|---|---|---|---|---|
| | 10-shot | 1-shot | 10-shot | 1-shot |
| Pin-Tuning | 54.97±9.05 | 35.87±3.36 | 87.45±3.63 | 72.48±1.61 |
| RECOG | **66.95±5.17** | **52.89±3.96** | **89.04±1.80** | **82.00±2.23** |

## D.5. $K$-shot Sensitivity Analysis

To examine the sensitivity of RECOG to the number of available support samples, we report the results under $\{1, 5, 10\}$-shot settings. We compare RECOG with Pin-Tuning, which is one of the strongest competing baselines in our experiments. The results on SIDER and Tox21 are summarized in Table 12. As the number of support samples increases, the performance of both methods generally improves, which is consistent with the common behavior of few-shot learning methods. More importantly, RECOG consistently outperforms Pin-Tuning across different shot settings on both datasets. These results indicate that RECOG remains effective under varying levels of data availability and can better exploit additional support samples when they become available.

## D.6. Evaluation with PR-AUC

In addition to ROC-AUC, we further report PR-AUC, which is particularly informative for molecular property prediction tasks with imbalanced label distributions. As shown in Table 13, RECOG consistently outperforms Pin-Tuning on Tox21 and SIDER under both 10-shot and 1-shot settings, with especially clear gains on Tox21 under the 1-shot setting. Together with the ROC-AUC results, these PR-AUC results indicate that RECOG demonstrates advantages not only in overall discrimination ability, but also in recognizing positive samples under imbalanced few-shot scenarios.

