# OpenReview forum: "ReCoG: Relational and Compact Context Graph Learning for Few-shot Molecular Property Prediction"
_ICML.cc/2026/Conference — ICML 2026 regular_

### Official Review · Reviewer_Nais · 2026-03-11

**Soundness:** 3
**Presentation:** 3
**Significance:** 3
**Originality:** 3
**Overall Recommendation:** 4
**Confidence:** 3

**Summary:**

The authors investigate an important problem in this work: few-shot molecular property prediction. Existing methods still face two critical challenges, namely insufficient modeling of structural contextual information and the learning of redundant auxiliary context. To address these issues, the authors propose a novel framework named RECOG, which is based on Relational and Compact Context Graph learning. The proposed RECOG consists of two key components: (1) a cross-attribute relational learning module, designed to better capture structural and relational contextual information; and (2) a context graph information bottleneck module, which adaptively suppresses irrelevant auxiliary signals. Extensive experiments on multiple datasets demonstrate that RECOG consistently outperforms state-of-the-art methods, validating its effectiveness.

**Compliance With Llm Reviewing Policy:**

Affirmed.

**Key Questions For Authors:**

See the weaknesses.

**Limitations:**

yes

**Strengths And Weaknesses:**

**Strengths**

1. The paper is clearly written and well-structured. The authors first present the theoretical motivation and then introduce the corresponding methodological design, making the overall framework easy to follow.

2. The proposed method is validated through extensive experiments on multiple datasets, and ablation studies further demonstrate the effectiveness of the individual components.

3. Moreover, the availability of the source code improves the reproducibility of the work.



**Weaknesses**

1. The proposed method adopts a bi-level optimization strategy during training, which may introduce additional training instability and incur extra computational overhead. The authors are encouraged to further discuss this issue.

2. In Table 1, a considerable amount of data is missing. The authors should consider completing these missing entries. If the corresponding results are not reported in the original papers, the authors may consider running these baselines on the relevant datasets to provide a more comprehensive comparison.

3. In Figure 5, the authors analyze the impact of the number of auxiliary tasks on the model performance. However, there exist several few-shot learning models [1][2] that are specifically designed to perform well in fewer-task scenarios. The authors may consider including comparisons with these methods or providing further discussion regarding their performance in such settings.

[1] Meta-learning with fewer tasks through task interpolation. In ICLR.

[2] Dual-level mixup for graph few-shot learning with fewer tasks. In WWW.

---

> ### Author Rebuttal · Authors · 2026-03-31
>
> Thanks for your constructive comments. We respond to these points below in a structured manner. The corresponding experiments, discussion, and references will be incorporated into the revised manuscript.
>
> **Q1**. The authors are encouraged to further discuss that “bi-level optimization strategy may introduce additional training instability and incur extra computational overhead”. \
> **A2**: We would like to clarify that bi-level optimization is a widely used training paradigm in few-shot and meta-learning methods, as it naturally models adaptation from the support set to the query set, and from meta-training tasks to meta-test tasks. Under this standard framework, ReCoG achieves the best performance among many competitive baselines, while also showing advantages in time efficiency. At the same time, such a strategy may introduce additional training instability and computational overhead. In fact, we have already noted in the conclusion that the relatively high variance on some datasets may be related to the stochastic episodic sampling strategy.\
> In the revised version, we will further enrich the discussion by noting that future improvements may come from introducing more structured training schedulers to better organize task sampling and context construction during meta-training, adopting more efficient gradient approximation strategies to replace standard MAML-style second-order differentiation, and exploring implicit-gradient or implicit-function-based optimization methods to further reduce memory and computational cost while improving training stability in bilevel optimization.
>
> **Q2**. Missing results in Table 1.\
> **A2**:  Following your advice, we further supplemented several sub-optimal missing baseline results where the source code was available and the rebuttal time permitted, including Pre-GNN, Meta-MGNN, Pre-ADKF-IFT, and Pre-PACIA. We would also like to clarify that most results reported in Table 1 were taken from the original papers or from published reproductions in authoritative venues. Overall, the newly supplemented results, together with the existing ones, further confirm the superiority of ReCoG over the baselines.
> |Method|Tox21-10|Tox21-1|SIDER-10|SIDER-1|MUV-10|MUV-1|ToxCast-10|ToxCast-1|PCBA-10|PCBA-1|
> |-|-|-|-|-|-|-|-|-|-|-|
> |Pre-GNN|82.14±0.08|81.68±0.09|73.96±0.08|73.24±0.12|67.14±1.58|64.51±1.45|73.68±0.74|72.90±0.84|76.79±0.45|75.27±0.49|
> |Meta-MGNN|82.94±0.10|82.13±0.13|75.43±0.21|73.36±0.32|68.99±1.84|65.54±2.13|76.27±0.57|72.43±0.85|72.58±0.34|72.51±0.35|
> |Pre-ADKF-IFT|86.06±0.35|80.97±0.48|70.95±0.60|62.16±1.03|95.74±0.37|67.25±3.87|76.22±0.13|71.13±1.15|80.21±0.27|76.62±0.73|
> |Pre-PACIA|86.40±0.27|84.35±0.14|83.97±0.22|80.70±0.28|73.43±1.96|69.26±2.35|76.22±0.73|75.09±0.95|75.36±0.30|70.16±1.33|
> |ReCoG|**93.52±1.47**|**91.01±2.03**|**93.98±1.53**|**90.06±1.17**|**95.94±1.37**|**81.96±0.42**|**88.85±1.39**|**86.05±1.74**|**82.57±1.42**|**82.39±1.64**|
>
> **Q3**. More few-shot baselines [1-2].\
> **A3**: Due to the time limit inthe rebuttal period, we made our best effort to reproduce these two methods and apply them to Tox21 and SIDER. The comparison results with ReCoG are shown below, further verifying the effectiveness of our method. Notably, both methods mainly rely on mixup-based data augmentation to improve model generalization, which may introduce chemically implausible or inaccurately labeled samples in molecular property prediction. In contrast, ReCoG enhances few-shot generalization through cross-property contextual relation learning rather than synthetic sample generation.\
> Moreover, we plan to include comparisons with these two important baselines in the appendix of the revised version.
> |Method|Tox21-10|SIDER-10|Tox21-1|SIDER-1|
> |-|-|-|-|-|
> |MLTI|78.95±0.17|81.57±1.72|65.49±2.45|65.85±1.48|
> |SMILE|76.23±0.69|71.80±0.79|60.69±1.24|56.49±1.49|
> |ReCoG|**93.52±1.47**|**93.98±1.53**|**91.01±2.03**|**90.06±1.17**|
>
> [1] Meta-learning with fewer tasks through task interpolation. In ICLR.\
> [2] Dual-level mixup for graph few-shot learning with fewer tasks. In WWW.

---

> > ### Author Rebuttal · Reviewer_Nais · 2026-04-02
> >
> > Thank you for the detailed response. I appreciate the additional experiments provided, which help strengthen the empirical validation of the work. The authors may consider including these results in the revised version of the paper. Accordingly, I have decided to maintain my positive score.

---

> > > ### Author Response · Authors · 2026-04-03
> > >
> > > Thank you very much for your positive and encouraging response. We truly appreciate your recognition of the additional experiments and are glad that they help strengthen the empirical validation of our work. We will make sure to incorporate these results into the revised version of the paper. We also noticed that the current status is marked as Partially resolved. If there are any remaining questions or concerns, we would be more than happy to continue the discussion and provide further clarification. Meanwhile, we would sincerely appreciate it if you could kindly reconsider your score in light of the additional evidence and clarifications provided in the rebuttal.

---

### Official Review · Reviewer_wEWh · 2026-03-12

**Soundness:** 3
**Presentation:** 3
**Significance:** 3
**Originality:** 3
**Overall Recommendation:** 4
**Confidence:** 4

**Summary:**

This paper proposes RECOG, a relational and compact context graph learning framework for few-shot molecular property prediction task. The RECOG aims to improve context-aware few-shot learning by addressing two challenges in existing approaches: insufficient modeling of cross-property relations and the presence of redundant auxiliary task information. To solve these issues, the authors introduce a cross-property relation learning (CPRL) module and a context graph information bottleneck (CGIB) module. Experiments on multiple MoleculeNet benchmarks show consistent improvements over existing few-shot molecular property prediction methods in both 1-shot and 10-shot settings.

**Compliance With Llm Reviewing Policy:**

Affirmed.

**Final Justification:**

In my initial review, I raised concerns regarding (1) the statistical justification for MSE-based relational learning under binary supervision, (2) the choice of Bernoulli gating over standard mutual information estimation, (3) limited hyperparameter analysis, and (4) the lack of evaluation on more challenging datasets. The authors' rebuttal addressed these concerns to a satisfactory degree. Although there are some minor concerns remain like the more detailed parameter studies, the contributions are solid overall and I am raising my recommendation to weak accept.

**Key Questions For Authors:**

1. Why is MSE chosen instead of a classification loss such as binary cross-entropy?

**Limitations:**

yes

**Strengths And Weaknesses:**

Strengths:
1. The paper is clearly motivated and the presentation is easy to follow.
2. The idea of using an information bottleneck mechanism to filter auxiliary task information is reasonable.
3. Consistent performance improvements across multiple datasets.

Weaknesses:
1. The design of the relation signal is overly simplistic. The cross-property relation signal is defined as the squared difference between two binary labels, which is not a common practice. Moreover, the relation predictor is trained using an MSE loss while the underlying variables are binary. Modeling the difference of Bernoulli variables with a Gaussian likelihood may not be statistically well justified, and using a classification-based objective like BCE might be more appropriate.
2. The information bottleneck formulation relies on a Bernoulli gating approximation. The reason of not using standard mutual information-based estimation requires additional justification and empirical validation.
3. The parameter studies are limited to the bottleneck coefficient $\beta$, while other important hyperparameters are not explored.
4. All experiments are conducted on datasets derived from MoleculeNet. While these benchmarks are widely used, the reported 1-shot performance on most datasets are already quite high, which implies the neccesity of introducing more challenging or diverse datasets like Meta-MolNet [1].

[1] Meta-MolNet: A Cross-Domain Benchmark for Few Examples Drug Discovery. TNNLS 2024

---

> ### Author Rebuttal · Authors · 2026-03-31
>
> Thanks for your valuable comments. We respond to these points below in a structured manner. The corresponding experiments, discussion, and references will be incorporated into the revised version.
>
> **Q1**. Why is MSE instead of a classification loss?\
> **A1**: We justify this design from both theoretical and empirical perspectives.\
> [Theoretical justification] Although the supervision is binary, our goal is not to perform standard classification, but to model a cross-property relation signal. From this perspective, the target can be naturally treated as a relation regression signal, and similar MSE-based designs under binary supervision have been adopted in prior relation learning studies [1,2].\
> [Empirical validation] We further compare **ReCoG-BCE**, **ReCoG**, and several strong baselines. The results show that ReCoG achieves the best overall performance, while ReCoG-BCE still outperforms many baselines, further validating the effectiveness of our relation learning objective. In addition, ReCoG consistently performs better than ReCoG-BCE, suggesting that MSE is more suitable for our relation modeling objective in practice.
> |Method|Tox21-10|SIDER-10|Tox21-1|SIDER-1|
> |-|-|-|-|-|
> |Pre-GS-Meta|86.91±0.41|85.08±0.54|86.46±0.55|84.45±0.26|
> |Pre-PACIA|86.40±0.27|83.97±0.22|84.35±0.14|80.70±0.28|
> |Pre-KRGTS|87.62±0.29|85.09±0.31|87.54±0.11|84.61±0.16|
> |Pin-Tuning|91.56±2.57|93.41±3.52|85.71±1.33|84.36±1.73|
> |ReCoG-BCE|91.50±1.66|91.79±0.86|88.79±3.48|87.71±2.76|
> |ReCoG|**93.52±1.47**|**93.98±1.53**|**91.01±2.03**|**90.06±1.17**|
>
> **Q2**. Reason for not using standard mutual information-based estimation requires additional justification and empirical validation.\
> **A2**: We further clarify our design choice from both theoretical and empirical perspectives.\
> [Theoretical perspective.] Our objective is not merely to estimate a mutual information value, but to perform node-wise selective filtering over the auxiliary context subgraph. Such keep-or-suppress behavior in graph data is naturally suited to a Bernoulli-style gating mechanism. This choice is also in line with prior graph information bottleneck studies, which commonly adopt stochastic masking, stochastic attention, or noise injection to realize tractable graph compression [3-4]. Moreover, since the environment and task subgraphs are designed to be complementary, explicitly strengthening their mutual-information coupling may instead cause redundant alignment, weaken contextual diversity, and degrade performance, as verified in Appendix D.3.\
> [Empirical perspective.] To further validate this point, we compare our **ReCoG(Bernolli)** vs. standard mutual information estimator MINE[5], i.e., **ReCoG-MINE**. The results show that ReCoG consistently outperforms ReCoG-MINE across datasets and few-shot settings, indicating that the ReCoG's Bernoulli-gated bottleneck is more suitable than standard MI estimation in our scenario.
> |Shot|Method|Tox21|SIDER|
> |-|-|-|-|
> |10|ReCoG-MINE|91.45±1.91|90.81±1.39|
> ||ReCoG(Bernolli)|**93.52±1.47**|**93.98±1.53**|
> |1|ReCoG-MINE|90.15±3.06|87.52±2.59|
> ||ReCoG(Bernolli)|**91.01±2.03**|**90.06±1.17**|
>
> **Q3**. Lack hyperparameters study.\
> **A3**: We further provide a parameter study on the Gumbel-Sigmoid temperature coefficient with {1.E-01, 2.E-01, 5.E-01,1.E+00, 2.E+00}. The results show that either overly small or overly large temperature values lead to performance degradation. Based on this observation, we choose 1.E+00 as the temperature coefficient in this paper.
> |Dataset|Shot|1.E-01|2.E-01|5.E-01|1.E+00|2.E+00|
> |-|-|-|-|-|-|-|
> |SIDER|10|89.40±3.07|92.62±4.39|92.12±0.24|**93.98±1.53**|90.13±1.05|
> ||1|85.44±1.78|85.98±1.65|87.48±1.77|**90.06±1.17**|88.37±3.21|
> |Tox21|10|89.37±1.67|90.00±1.56|91.25±1.50|**93.52±1.47**|91.47±1.23|
> ||1|88.62±1.55|89.17±1.63|89.58±1.91|**91.01±2.03**|88.69±4.28|
>
> **Q4**. More 1-shot challenging or diverse datasets like Meta-MolNet[6].\
> **A4**: We further evaluate ReCoG on two more challenging Meta-MolNet[6] datasets, JNK3 and HIV, which remain difficult even under 1-shot setting. Compared with the sota baseline Meta-GAT[7] reported in Meta-MolNet, ReCoG achieves better performance on both datasets, further demonstrating its generalization ability and superiority.
> |Method|JNK3|HIV|
> |-|-|-|
> |Meta-GAT|86.45±1.63|87.41±0.15|
> |ReCoG|**92.44±4.28**|**92.59±0.91**|
>
> [1] Learning to Compare: Relation Network for Few-Shot Learning, CVPR 2018.\
> [2] PARN: Position-Aware Relation Networks for Few-Shot Learning, ICCV2019.\
> [3] Interpretable and Generalizable Graph Learning via Stochastic Attention Mechanism, ICML 2022.\
> [4] Contrastive Graph Structure Learning via Information Bottleneck for Recommendation, NeurIPS
> 2022.\
> [5] Mutual information neural estimation, ICML2018.\
> [6] Meta-MolNet: A Cross-Domain Benchmark for Few Examples Drug Discovery. TNNLS 2024.\
> [7] Meta Learning With Graph Attention Networks for Low-Data Drug Discovery. TNNLS 2023.

---

> > ### Author Rebuttal · Reviewer_wEWh · 2026-04-03
> >
> > Thank you for the detailed rebuttal. The authors have adequately addressed my main concerns. I am raising my score to 4 accordingly.

---

> > > ### Author Response · Authors · 2026-04-03
> > >
> > > Thank you very much for your thoughtful follow-up and for raising your score. We truly appreciate your recognition that the rebuttal has adequately addressed your main concerns. Your constructive feedback has been very helpful in improving our work, and we will make sure to incorporate the relevant clarifications into the revised version.

---

### Official Review · Reviewer_X7bs · 2026-03-12

**Soundness:** 3
**Presentation:** 3
**Significance:** 3
**Originality:** 3
**Overall Recommendation:** 4
**Confidence:** 4

**Summary:**

This paper proposes a method for few-shot molecular property prediction that aims to address two challenges: insufficient structural context modeling and redundant auxiliary context learning. The method is theoretically motivated and integrates relational context graphs with an information bottleneck objective. Experimental results show that the model outperforms previous state-of-the-art methods on several MoleculeNet datasets, including Tox21, SIDER, and ToxCast.

**Compliance With Llm Reviewing Policy:**

Affirmed.

**Key Questions For Authors:**

1. Is the proposed few-shot learning framework specific to molecular property prediction, or could it generalize to other domains and tasks?
2. How sensitive is the method to the number of available samples in the few-shot setting?
3. Would ReCoG still outperform baselines with substantially different molecular encoders?

**Limitations:**

Although the paper does acknowledge variance caused by stochastic episodic sampling, which is a useful point, the discussion of limitations should be broader.

**Strengths And Weaknesses:**

+ Strengths:
- The paper proposes a new approach for few-shot learning that addresses two challenges in molecular property prediction: insufficient structural context modeling and redundant auxiliary context learning.
- The model achieves strong performance on several benchmark datasets.
- The ablations are useful.

+ Weaknesses:
- The proposed method does not appear to be specifically designed for molecular property prediction. It would be helpful to clarify whether the approach can generalize to other domains beyond molecular tasks.
- It is not clear how much data the method needs in order to perform well in the few-shot setting.
- Evaluating drug design tasks using only binary classification and reporting only ROC-AUC is not sufficient; PR-AUC would be more appropriate and informative.

---

> ### Author Rebuttal · Authors · 2026-03-31
>
> Thanks for your thoughtful comments. We respond to these points below in a structured manner. The corresponding experiments, discussion, and references will be incorporated into the revised manuscript.
>
> **Q1.** Is the proposed few-shot learning framework specific to molecular property prediction, or could it generalize to other domains and tasks?\
> **A1**: We would like to clarify that our framework is developed specifically for molecular property prediction, from the overall motivation and theoretical analysis to the method design. The core idea of our method is to learn a relational and compact molecule-property context graph, which is instantiated through a cross-property relational learning module and a molecule-property context graph information bottleneck module. These designs are closely tied to the characteristics of molecular property prediction and the underlying molecule-property relational structure. Moreover, we are happy to show the potential of the proposed framework for extension to other domains and tasks that also involve structured relational context. In such cases, the general idea of learning a relational and compact task context graph may still be applicable, although the specific encoder, context construction, and relation learning modules would need to be redesigned according to target domains.
>
> **Q2&Q4**. How sensitive is the method to the number of available samples in few-shot setting? It is not clear how much data the method needs in order to perform well. \
> **A2&A4**: ReCoG is slightly sensitive to the number of available samples, which is a common characteristic of few-shot learning methods. Here, we provide results of ReCoG and sub-optimal baseline Pin-Tuning under 1\5\10-shot settings, which show that the performance generally improves as more support samples become available. Meanwhile, ReCoG consistently outperforms the strongest baselines across different few-shot settings, demonstrating its effectiveness under varying levels of data availability.
> |Dataset|SIDER|||Tox21|||
> |-|-|-|-|-|-|-|
> |Shot|1|5|10|1|5|10|
> |Pin-Tuning|84.36±1.73|92.02±3.01|93.41±3.52|85.71±1.33|90.95±2.33|91.56±2.57|
> |ReCoG|**90.06±1.17**|**92.87±0.78**|**93.98±1.53**|**91.01±2.03**|**91.69±0.64**|**93.52±1.47**|
>
> **Q3**. Would ReCoG still outperform baselines with different molecular encoders?\
> **A3**: Yes, we've tried to replace the encoder in ReCoG with GCN and GraphSAGE, and found that ReCoG still outperforms some baselines with GIN on some tasks. At the same time, the performance of these two variants is worse than that of ReCoG with GIN, which is also consistent with the conclusion in Pre-GNN that GIN is the most expressive encoder among these candidates. Moreover, we would like to clarify that the ReCoG results reported in our paper are obtained under the same molecular encoder architecture as many strong baselines, including Pre-GS-Meta, Pre-PACIA, Pre-KRGTS, and Pin-Tuning. Therefore, the reported improvements provide fair and effective evidence for the superiority of ReCoG itself, rather than stemming from encoder differences.
> |Dataset|Shot|Method|GCN|GraphSAGE|GIN|
> |-|-|-|-|-|-|
> |SIDER|10|Pin-Tuning|86.74±0.70|89.33±3.54|93.41±3.52|
> |||ReCoG|**90.93±1.42**|**90.35±1.24**|**93.98±1.53**|
> ||1|Pin-Tuning|79.67±1.46|79.05±3.52|84.36±1.73|
> |||ReCoG|**87.85±1.65**|**86.74±2.10**|**90.06±1.17**|
> |Tox21|10|Pin-Tuning|90.12±1.84|90.90±1.00|91.56±2.57|
> |||ReCoG|**92.05±3.07**|**91.74±2.73**|**93.52±1.47**|
> ||1|Pin-Tuning|84.66±1.18|85.62±1.27|85.71±1.33|
> |||ReCoG|**86.73±1.53**|**86.50±1.18**|**91.01±2.03**|
>
> **Q5**: Evaluating drug design tasks using only binary classification and reporting only ROC-AUC is not sufficient; PR-AUC would be more appropriate and informative.\
> **A5**: Thanks for the valuable suggestion regarding the metric PR-AUC. Accordingly, we further evaluate the PR-AUC of the model at its best ROC-AUC checkpoint and report the comparison with the sub-optimal method Pin-Tuning on Tox21 and SIDER. The additional results further verify the effectiveness of ReCoG.
> |Metric|Shot|Method|Tox21|SIDER|
> |-|-|-|-|-|
> |PR-AUC|10|Pin-Tuning|54.97±9.05|87.45±3.63|
> |||ReCoG|**66.95±5.17**|**89.04±1.80**|
> ||1|Pin-Tuning|35.87±3.36|72.48±1.61|
> |||ReCoG|**52.89±3.96**|**82.00±2.23**|
>
> **Q6**. Although the paper does acknowledge variance caused by stochastic episodic sampling, which is a useful point, the discussion of limitations should be broader.\
> **A6**: In revised version, we will further enrich the discussion by noting that future improvements may come from introducing more structured training schedulers to better organize task sampling and context construction during meta-training, adopting more efficient gradient approximation strategies to replace standard MAML-style second-order differentiation, and exploring implicit-gradient or implicit-function-based optimization methods to further reduce memory and computational cost while improving training stability in bilevel optimization.

---

> > ### Author Rebuttal · Reviewer_X7bs · 2026-04-02
> >
> > Although most of my concerns have been addressed, I will keep my rating unchanged, as, in my view, it reflects the extent to which the contributions of this work align with the standard expected at ICML.

---

> > > ### Author Response · Authors · 2026-04-03
> > >
> > > Thank you very much for the time and effort you have devoted to reviewing our paper. We also sincerely appreciate your recognition of our rebuttal and are glad that most of your concerns have been addressed. While we fully respect your current assessment, we would sincerely appreciate it if you could kindly reconsider the score in light of the strengthened empirical validation and clarifications provided in the rebuttal.

---

### Decision · Program_Chairs · 2026-04-30

**Decision:**

Accept (regular)

**Comment:**

This paper proposes ReCoG, a framework for few-shot molecular property prediction (FSMPP) that introduces two core modules: a cross-property relation learning (CPRL) module and a context graph information bottleneck (CGIB) module. All three reviewers assigned Weak Accept, placing this submission at the borderline. Scores were largely maintained after the rebuttal phase.

### Strengths:

The paper presents a reasonable motivation, and the observation that increasing the number of auxiliary tasks does not always improve performance provides a sensible starting point. ReCoG shows consistent improvements over baselines across five MoleculeNet benchmarks under both 10-shot and 1-shot settings. The rebuttal provided additional experiments on encoder robustness, PR-AUC evaluation, Meta-MolNet benchmarks, and hyperparameter sensitivity, which addressed several reviewer concerns.

### Weaknesses and Additional AC Assessment:

Despite the unanimous Weak Accept, I have identified significant concerns regarding the technical novelty of both core modules that were not sufficiently examined during the review process.

First, the design of the relation signal in the CPRL module is overly simplistic. The cross-property relation signal is defined as the squared difference between two binary labels, and the relation predictor is trained with MSE loss. This effectively amounts to adding a single auxiliary regression loss to the training objective. While Reviewer wEWh raised this concern, the rebuttal addressed it only with an empirical comparison showing that MSE outperforms BCE, which does not resolve the underlying concern about the limited methodological depth of this module.

Second, and more critically, the CGIB module closely follows the established pipeline of existing graph information bottleneck methods—specifically, the Bernoulli gating with Gumbel-Sigmoid relaxation and Gaussian noise injection scheme introduced in VGIB (Yu et al., CVPR 2022) and GSAT (Miao et al., ICML 2022)—applied to a conditional setting.
Notably, this conditional graph information bottleneck formulation has already been proposed in Lee et al. (ICML 2023) [1]. The methodological parallels are substantial: (1) decomposing the conditional mutual information via the chain rule into two terms, (2) the design choice of omitting one of the decomposed terms based on empirical observation that it degrades performance, and (3) deriving a variational upper bound through noise injection into node representations with Bernoulli gating and Gumbel-Sigmoid reparameterization. Despite these strong similarities, the present paper does not cite this prior work at all. This omission results in an overclaimed novelty for the CGIB module and constitutes a significant gap in the coverage of related work.

Taken together, the two core modules reduce to (1) a trivial auxiliary loss addition and (2) a straightforward application of existing conditional graph IB techniques to the FSMPP domain. While the empirical results are strong, the contribution of this paper is effectively limited to a domain-specific combination of known techniques with thorough experimental validation.

Despite the unanimous Weak Accept from all three reviewers, the insufficient novelty of both core modules and the omission of the most closely related prior work on conditional graph information bottleneck represent critical issues that were overlooked during the review process. I encourage the authors to (1) develop a more principled and sophisticated design for the cross-property relation signal, and (2) clearly articulate the technical distinctions from prior work on conditional graph information bottleneck before resubmission.

Reference:
[1] Lee et al, “Conditional Graph Information Bottleneck for Molecular Relational Learning.” In Proceedings of the 40th International Conference on Machine Learning (ICML), 2023.​​​​​​​​​​​​​​​​